# Consistent and correctable bias in metagenomic sequencing experiments

**Michael R McLaren[1], Amy D Willis[2], Benjamin J Callahan[1,3]***

[1]Department of Population Health and Pathobiology, North Carolina State University, Raleigh, United States; [2]Department of Biostatistics, University of Washington, Seattle, United States; [3]Bioinformatics Research Center, North Carolina State University, Raleigh, United States

**Abstract** Marker-gene and metagenomic sequencing have profoundly expanded our ability to measure biological communities. But the measurements they provide differ from the truth, often dramatically, because these experiments are biased toward detecting some taxa over others. This experimental bias makes the taxon or gene abundances measured by different protocols quantitatively incomparable and can lead to spurious biological conclusions. We propose a mathematical model for how bias distorts community measurements based on the properties of real experiments. We validate this model with 16S rRNA gene and shotgun metagenomics data from defined bacterial communities. Our model better fits the experimental data despite being simpler than previous models. We illustrate how our model can be used to evaluate protocols, to understand the effect of bias on downstream statistical analyses, and to measure and correct bias given suitable calibration controls. These results illuminate new avenues toward truly quantitative and reproducible metagenomics measurements.

DOI: https://doi.org/10.7554/eLife.46923.001

**\*For correspondence:**
benjamin.j.callahan@gmail.com

**Competing interests:** The authors declare that no competing interests exist.

## Introduction

Marker-gene and metagenomic sequencing (jointly, MGS) have transformed the study of biological communities. Extracting and sequencing total DNA from a community can identify thousands of taxa along with their genes and potential functions, while sequencing a phylogenetic marker gene (e.g. 16S rRNA) can quantify taxon abundances (*Li, 2015*; *Quince et al., 2017*). MGS measurements of microbial communities are yielding fundamental new insights into the structure and dynamics of microbial ecosystems and the roles of microbes as drivers of host and ecosystem health (*Zeevi et al., 2015*; *Graham et al., 2016*; *Knight et al., 2017*; *Callahan et al., 2017*; *Lehman et al., 2015*). Applications of MGS, often under the alternative terms eDNA sequencing or metabarcoding, increasingly extend beyond microbes to the measurement and monitoring of plants, insects, and vertebrates (*Bell et al., 2019*; *Krehenwinkel et al., 2017*; *Thomas et al., 2016*). MGS methods are now being adopted in fields ranging from food safety (*Cocolin et al., 2018*) to wastewater remediation (*Rosso et al., 2018*) to forensics (*Metcalf et al., 2017*) along with biology and medicine. Unfortunately, however, the community compositions measured by MGS are wrong.

MGS measurements are *biased*: The measured relative abundances of the taxa and genes in the sample are systematically distorted from their true values (*Brooks, 2016*; *Sinha et al., 2017*). Bias arises because each step in an experimental MGS workflow preferentially measures (i.e. preserves, extracts, amplifies, sequences, or bioinformatically identifies) some taxa over others (*Brooks, 2016*; *Hugerth and Andersson, 2017*; *Pollock et al., 2018*). For example, bacterial species differ in how easily they are lysed and therefore how much DNA they yield during DNA extraction (*Morgan et al., 2010*; *Costea et al., 2017*), and they differ in their number of 16S rRNA gene copies and thus how much PCR product we expect to obtain per cell (*Kembel et al., 2012*). Most sources of bias are

protocol-dependent: Different PCR primers preferentially amplify different sets of taxa (*Sipos et al., 2007*), different extraction protocols can produce 10-fold or greater differences in the measured proportion of a taxon from the same sample (*Costea et al., 2017*), and almost every choice in an MGS experiment has been implicated as contributing to bias (*D'Amore et al., 2016*; *Hugerth and Andersson, 2017*; *Sinha et al., 2017*; *Pollock et al., 2018*). Every MGS experiment is biased to some degree, and measurements from different protocols are quantitatively incomparable (*Nayfach and Pollard, 2016*; *Hiergeist et al., 2016*; *Mallick et al., 2017*; *Sinha et al., 2017*; *Gibbons et al., 2018*).

The biases of MGS protocols and the error those biases introduce remain unknown. Thus we do not know whether the measured taxonomic or gene compositions derived from MGS are accurate, or to what extent the biological conclusions derived from them are valid. It is common to assume that conclusions drawn from measurements using the same protocol are robust to MGS bias. But simulated examples have shown that bias can lead to qualitatively incorrect conclusions about which taxa dominate different samples (*Kembel et al., 2012*; *Edgar, 2017*), which ecosystems are more similar (*Kembel et al., 2012*), and which taxa are associated with a given disease (*Brooks, 2016*). Furthermore, variation in bias limits our ability to make the direct comparisons between results from different experiments that are central to the scientific process. It has been suggested that these issues would be circumvented if bias were the same in every experiment, leading to a number of efforts to define and promulgate standardized MGS protocols (*Gilbert et al., 2014*; *Costea et al., 2017*). However, methodological standardization has several limitations. For example, it can be overly restrictive given the variety of ecosystems and biological questions where MGS methods are applied as well as the continual advance in technology, and unmeasured technical variability can introduce experiment-specific biases into nominally standardized methods (*Yeh et al., 2018*). More important, standardized protocols remain biased and thus still do not provide accurate measurements of the underlying communities.

Current attempts to counter bias are limited and of unknown efficacy because of our poor understanding of how bias across the full experimental workflow distorts MGS measurements. Hundreds of published studies compare MGS measurements of defined samples to their expected composition in an effort to characterize the bias of the given protocol (many cited in *Hugerth and Andersson, 2017*; *Pollock et al., 2018*). But this approach has limited value so long as we do not know how the error observed in one sample translates to differently composed samples. If we understood how bias acts across samples we might be able to estimate the effect of bias from measurements of samples of defined composition and use those estimates to calibrate measurements of samples of interest to their true values (*Thomas et al., 2016*; *Hardwick et al., 2017*). Alternatively, natural communities measured by multiple experiments could be used to calibrate measurements between experiments using different protocols. A quantitative understanding of how bias distorts MGS measurements would also elucidate how statistical analyses and diagnostics are affected by bias and suggest more robust alternatives.

Here we propose and test a mathematical model of how bias distorts taxonomic compositions measured by MGS from their true values. In our model, bias manifests as a multiplication of the true relative abundances by taxon- and protocol-specific factors that are constant across samples of varying compositions. We validate key components of this model, including that bias acts independently on each taxon in a sample, in 16S rRNA gene and shotgun metagenomic sequencing data from bacterial communities of defined composition. We use our proposed model to quantify bias, to partition bias into steps such as DNA extraction and PCR amplification, and to reason about the effects of bias on downstream statistical analyses. Finally, we demonstrate how this model can be used to correct biased MGS measurements when suitable controls are available.

## Results

### A mathematical model of MGS bias

Consider a marker-gene or metagenomic sequencing (MGS) experiment as a multi-step transformation that takes as input biological material and provides as output the taxonomic profile corresponding to each sample—the set of measured taxa and their associated relative abundances (*Figure 1A*). Each step introduces systematic and random errors that cumulatively lead to error in the measured

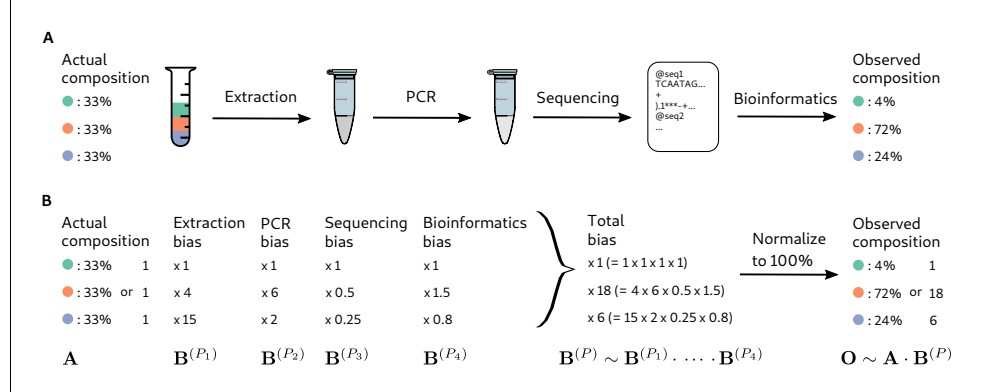

**Figure 1.** Bias arises throughout an MGS workflow, creating systematic error between the observed and actual compositions. Panel **A** illustrates a hypothetical marker-gene measurement of an even mixture of three taxa. The observed composition differs from the actual composition due to the bias at each step in the workflow. Panel **B** illustrates our mathematical model of bias, in which bias multiplies across steps to create the bias for the MGS protocol as a whole.

DOI: https://doi.org/10.7554/eLife.46923.002

taxonomic profiles. *Bias* is a particular, ubiquitous form of systematic error that arises from the different efficiencies with which various taxa are measured (i.e. preserved, extracted, amplified, sequenced, or bioinformatically identified and quantified) at each step.

Many bias mechanisms are thought to act multiplicatively on the taxon abundances, at least to first approximation. For instance, the DNA concentration of a taxon after DNA extraction equals its initial cell concentration multiplied by its DNA yield per cell. This per-cell yield indicates the efficiency of extraction for the taxon, which is is expected to depend on factors such as genome size and the structure of the taxon's cell wall (*Morgan et al., 2010*). Therefore, we expect extraction efficiencies to vary among taxa, but be approximately constant for any specific taxon across samples treated with the same protocol. Other multiplicative sources of bias include variation in PCR binding and amplification efficiencies (*Wagner et al., 1994*; *Suzuki and Giovannoni, 1996*; *Polz and Cavanaugh, 1998*; *Edgar, 2017*) and in marker-gene copy number (*Kembel et al., 2012*).

Inspired by these observations, we propose that at every step in an MGS experiment, the output abundances of individual taxa differ from the input abundances by taxon-specific multiplicative factors (*Figure 1*), which we refer to as the *measurement efficiencies* in that step. The measurement efficiencies are determined by the interaction between the experimental protocol and the biological/chemical/physical/informatic state of each taxon in that step, and are therefore independent of the composition of the sample. Typical MGS experiments only measure relative rather than absolute abundances (*Gloor et al., 2017*), and the change in the *relative abundances* during a step depend only on the *relative efficiencies*. This yields the following mathematical model of bias (*Figure 1*): The relative abundances measured in an MGS experiment are equal to the input relative abundances multiplied by taxon-specific but composition-independent factors (the relative efficiencies) at every step.

The mathematical accounting of bias is simplified by the use of *compositional vectors*: vectors for which only the ratios among elements carry meaning. The relative abundances and relative efficiencies can be described as compositional vectors with $K$ non-negative elements, where $K$ is the number of possible taxa. Two vectors $\mathbf{X}$ and $\mathbf{Y}$ are *compositionally equivalent*, denoted $\mathbf{X} \sim \mathbf{Y}$, if $\mathbf{X} = a\mathbf{Y}$ for some positive constant $a$ because the ratios among the elements of $\mathbf{X}$ and $\mathbf{Y}$ are the same: $X_i/X_j = aY_i/aY_j$ (*Barceló-Vidal et al., 2001*). A compositional vector $\mathbf{X}$ of relative abundances can be converted to proportions, which we denote $Pr(\mathbf{X})$, by dividing the taxon abundances by their sum, $Pr(\mathbf{X}) = \mathbf{X}/\sum_i X_i$, without changing its meaning in terms of the ratios among taxa. For example, the vector of observed proportions in *Figure 1* of (4%, 72%, 24%) is compositionally equivalent to the vector (1, 18, 6) obtained by dividing all abundances by that of the first taxon.

For a given sample, let $\mathbf{A}$ be the vector of actual relative abundances and $\mathbf{O}$ be the vector of observed (measured) relative abundances. Subscripts denote specific taxa; for example $A_i$ is the

relative abundance of taxon $i$. Similarly, let $\mathbf{B}^{(P_l)}$ be the vector of the relative efficiencies of each taxon at step $l$ in Protocol $P$. (Interactions between steps are allowed; see Appendix 1.) Our model of bias can be stated mathematically as

$$\mathbf{O} \sim \mathbf{A} \cdot \mathbf{B}^{(P_1)} \cdot \mathbf{B}^{(P_2)} \cdot \cdots \cdot \mathbf{B}^{(P_L)}, \tag{1}$$

where $\cdot$ denotes element-wise multiplication of two vectors (*Figure 1*). We define the bias of Protocol $P$ by the product over all steps, $\mathbf{B}^{(P)} \sim \mathbf{B}^{(P_1)} \cdot \mathbf{B}^{(P_2)} \cdot \cdots \cdot \mathbf{B}^{(P_L)}$. The observed composition is then simply the actual composition multiplied by the protocol's bias,

$$\mathbf{O} \sim \mathbf{A} \cdot \mathbf{B}^{(P)}. \tag{2}$$

When considering samples measured by the same protocol, we will drop the superscript $P$ and simply refer to the total protocol bias as $\mathbf{B}$.

From *Equation 2* we see that the ratio between the observed relative abundances of any two taxa $i$ and $j$ is

$$\frac{O_i}{O_j} = \frac{A_i B_i}{A_j B_j}, \tag{3}$$

and the observed proportion of taxon $i$ is

$$Pr(\mathbf{O})_i = \frac{O_i}{\sum_{j=1}^{K} O_j} = \frac{Pr(\mathbf{A})_i B_i}{\sum_{j=1}^{K} Pr(\mathbf{A})_j B_j}. \tag{4}$$

The denominator, $\sum_{j=1}^{K} Pr(\mathbf{A})_j B_j$, is the *sample mean efficiency*—the average efficiency of the sampled individuals.

The systematic error in the measured composition under our model is $\mathbf{O}/\mathbf{A} \sim \mathbf{B}$, where $/$ denotes element-wise division and is referred to as the compositional difference (*Aitchison, 1992*). The compositional difference unites the experimental notion of bias—variation in the efficiencies with which different taxa are measured—with the statistical notion of bias—the difference between the expected value of an estimate and the true value—with the understanding we are considering the compositional difference rather than the conventional Euclidean difference between compositions.

## Properties and implications of the model

### Bias is fully described by the relative efficiencies of the total workflow

The bias of individual steps only influences the measurement error through their product, $\mathbf{B}^{(P)}$. Consequently, knowledge of the total protocol bias is sufficient to determine how bias affects the measured taxonomic profiles even if the biases of the individual protocol steps (the $\mathbf{B}^{(P_l)}$) remain unknown. The bias $\mathbf{B}^{(P)}$ has just $K-1$ parameters, denoting the relative efficiencies with which the $K$ taxa of interest are measured by the protocol as a whole, and fully describes the effect of bias on measurements of those $K$ taxa in all samples.

### Systematic error in taxon ratios, but not in taxon proportions, is independent of sample composition

The fold-error in the observed ratios of the abundances of taxon $i$ and taxon $j$ relative to the actual ratio in their abundances is $(O_i/O_j)/(A_i/A_j) = B_i/B_j$ (*Equation 3*). This error depends only on the ratio between the total protocol efficiencies of taxon $i$ and taxon $j$ and is independent of the rest of the sample. Critically, this means that the systematic error in taxon ratios caused by bias will remain the same in samples of varying composition.

In contrast, the error in the proportion of a taxon depends on the sample composition. The fold-error in the observed proportion of taxon $i$ relative to its actual proportion is

$$\frac{Pr(\mathbf{O})_i}{Pr(\mathbf{A})_i} = \frac{B_i}{\sum_{j=1}^{K} Pr(\mathbf{A})_j B_j}. \tag{5}$$

This error depends on the sample mean efficiency $\sum_j Pr(\mathbf{A})_j B_j$ and thus depends on the proportions

of all the other taxa in the sample. Intuitively, bias leads to over-estimation of taxa that are more easily measured than the community average in the given sample. As a result, the same taxon can be over-estimated in samples dominated by low-efficiency taxa and under-estimated in samples dominated by high-efficiency taxa.

To illustrate, we consider the hypothetical measurement of a second community sample (Sample $S_2$ in *Figure 2*) alongside that of the even sample from *Figure 1* (Sample $S_1$ in *Figure 2*). The dominance of the low-efficiency Taxon 1 in Sample $S_2$ substantially lowers its sample mean efficiency compared to the even-mixture Sample $S_1$, changing the fold-error in all taxon proportions. In particular, Taxon 3 changes from having a lower-than-average efficiency in Sample $S_1$ to a higher-than-average efficiency in Sample $S_2$. As a result, its observed proportion is lower than its actual proportion in Sample $S_1$, but higher than its actual proportion in Sample $S_2$! Yet the fold-error in the ratios among taxa is identical in both samples and equal to the bias (*Figure 2*, bottom row).

## Analyses based on fold-changes in taxon ratios are insensitive to bias, while analyses based on taxon proportions can give spurious results

Although it is widely understood that bias distorts individual community profiles, it is often thought to effectively 'cancel out' when analyzing the differences between samples that have been measured by the same protocol. Unfortunately, simulating measurement under our model easily provides examples where common analyses give qualitatively incorrect results. In *Figure 2*, for example, bias causes the uneven Sample $S_2$ to appear to have a more even distribution of taxa than the perfectly even Sample $S_1$. As a result, any analysis of alpha diversity that incorporates evenness (e.g. the Shannon or Inverse Simpson indices) will incorrectly conclude that Sample $S_2$ is more diverse. The previous section provides a general explanation as to why, for many analyses, bias does not simply cancel: The underlying statistics are functions of the individual taxon proportions, the error of which varies inconsistently across samples. Consequently, proportion-based analyses can lead to qualitatively incorrect conclusions. As a further example, the actual proportion of Taxon 3 decreases from Sample $S_1$ to Sample $S_2$ in *Figure 2*, but the measured proportion increases!

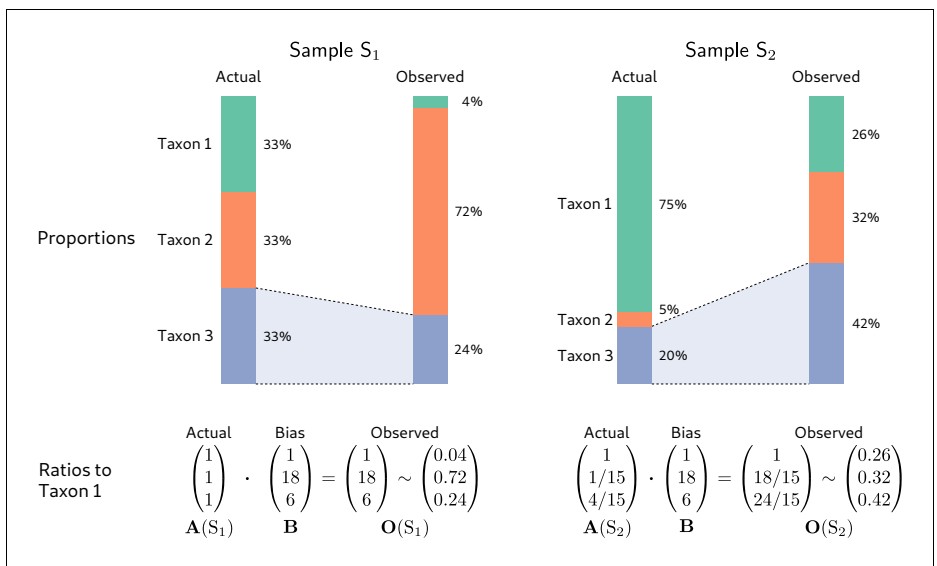

**Figure 2.** Consistent multiplicative bias causes systematic error in taxon ratios, but not taxon proportions, that is independent of sample composition. The even community from *Figure 1* and a second community containing the same three taxa in different proportions are measured by a common MGS protocol. Measurements of both samples are subject to the same bias, but the magnitude and direction of error in the taxon proportions depends on the underlying composition (top row). In contrast, when the relative abundances and bias are both viewed as ratios to a fixed taxon (here, Taxon 1), the consistent action of bias across samples is apparent (bottom row).
DOI: https://doi.org/10.7554/eLife.46923.003

In contrast, the fold-error in taxon ratios is independent of sample composition, and fold-changes in taxon ratios across samples are insensitive to bias. Consider the fold-change in the ratio of a pair of taxa $i$ and $j$ between two samples $s$ and $t$. Following *Equation 3*, the observed change is

$$\frac{O_i(s)}{O_j(s)} \Big/ \frac{O_i(t)}{O_j(t)} = \frac{A_i(s)B_i}{A_j(s)B_j} \Big/ \frac{A_i(t)B_i}{A_j(t)B_j} = \frac{A_i(s)}{A_j(s)} \Big/ \frac{A_i(t)}{A_j(t)}, \tag{6}$$

and thus equals the true change. That is, *the fold-change in taxon ratios between samples is invariant to bias*. More generally, the compositional difference between samples is invariant to multiplication by a fixed vector (*Aitchison, 1992*) and thus to bias,

$$\mathbf{O}(s) / \mathbf{O}(t) \sim (\mathbf{A}(s) \cdot \mathbf{B}) / (\mathbf{A}(t) \cdot \mathbf{B}) \sim \mathbf{A}(s) / \mathbf{A}(t). \tag{7}$$

Returning to the samples in *Figure 2*, the actual and observed ratios of Taxon 2 to Taxon 1 both change by the same factor of 1/15 from Sample $\mathrm{S}_1$ to Sample $\mathrm{S}_2$, and the actual and observed compositional difference between samples is (1, 1/15, 4/15). *Equation 7* shows that any analysis that depends only on the compositional differences between samples will be invariant to bias under our model.

## The systematic difference between measurements from different protocols is given by the difference in their biases

Consider Protocol $P$ with bias $\mathbf{B}^{(P)}$ and reference Protocol $R$ with bias $\mathbf{B}^{(R)}$. If both protocols measure the same sample with actual composition $\mathbf{A}$, the compositional difference between their measurements is

$$\mathbf{O}^{(P)} / \mathbf{O}^{(R)} \sim \mathbf{A} \cdot \mathbf{B}^{(P)} / (\mathbf{A} \cdot \mathbf{B}^{(R)}) = \mathbf{B}^{(P)} / \mathbf{B}^{(R)}. \tag{8}$$

The actual composition drops from the equations and the difference in their measurements is simply the compositional difference in the biases of each protocol, which we refer to as the differential bias $\mathbf{B}^{(P/R)} \equiv \mathbf{B}^{(P)}/\mathbf{B}^{(R)}$ of Protocol $P$ relative to the reference Protocol $R$. Measurements on common samples are related to one another by $\mathbf{O}^{(P)} \sim \mathbf{O}^{(R)} \cdot \mathbf{B}^{(P/R)}$, independent of the actual composition of the sample. Usefully, differential bias is mathematically equivalent to bias if we consider the 'reference' compositions measured by Protocol $R$ as the truth.

## Estimates of bias from control samples can be used to correct measurements of other samples

The consistency of bias across samples makes it possible to estimate bias from samples of known composition, referred to as *calibration controls*, and to use that estimate $\hat{\mathbf{B}}$ to *calibrate*, or remove the bias from, measurements of other samples with unknown compositions. A point estimate of the bias of $K$ taxa present with known relative abundances in control sample $c$ is given by the compositional difference between the observed and actual compositions, $\hat{\mathbf{B}} \sim \mathbf{O}(c)/\mathbf{A}(c)$. In Materials and methods, we describe a general method for estimating bias from multiple controls by maximizing the explained compositional error in the control measurements. Measurements from controls containing different taxa can be combined into a single estimate of bias provided that the controls have sufficient taxonomic overlap (Appendix 2).

Once bias has been estimated for a set of taxa, it can be used to calibrate the relative abundances of those taxa in an unknown sample. Letting $\mathbf{O}$ denote the measured relative abundances for these taxa, the estimate $\hat{\mathbf{A}}$ of the actual relative abundances is

$$\hat{\mathbf{A}} \sim \mathbf{O} / \hat{\mathbf{B}}. \tag{9}$$

That is, the calibrated abundances are found by compositionally subtracting the estimated bias from the original measurement. Through its use of compositional vectors, *Equation 9* automatically accounts for differences in composition between the controls and the target sample. Calibrated estimates of the true taxon proportions are obtained by normalizing the elements of $\hat{\mathbf{A}}$ to sum to 1.

An alternative form of calibration we call *reference calibration* can be performed using control samples whose true composition is unknown but that have been measured by a reference Protocol

$R$. Estimation and calibration proceed as before but with the control composition $\mathbf{A}(c)$ replaced by the reference measurement $\mathbf{O}^{(R)}(c)$. In this case, the calibrated composition is an estimate of the measurement we would expect if the target sample had been measured by the reference protocol.

## Testing the model with mock communities

We tested our model of bias in data from two studies, *Brooks et al. (2015)* and *Costea et al. (2017)*, that evaluated the bias of marker-gene and shotgun metagenomic sequencing, respectively, using mock microbial communities in samples of varying composition.

### Marker-gene sequencing of even mixtures of various bacterial taxa

*Brooks et al. (2015)* generated taxonomic profiles from 71 samples of 58 unique mock communities by amplicon sequencing of the V1-V3 region of the 16S rRNA gene. Each unique mock community consisted of an even mixture of between two and seven bacterial taxa. Each sample was measured in three experiments employing a common experimental workflow, but beginning from different starting points: even mixtures of cells, of extracted DNA, and of PCR product. The authors reported large systematic errors in the taxon proportions measured from the cell and DNA mixtures, which they explained in part by a highly parameterized linear model with many interaction terms. Here we re-analyze the data from this study in order to evaluate our model of bias and its performance relative to alternatives.

The proportions measured from the cell-mixture mock communities differed greatly from the expected even proportions of each taxon (*Figure 3A*). The ratios between pairs of taxa also diverged sharply from the ratio of 1 expected in these even mixtures (*Figure 3D*). However, and as predicted by our model (see Properties and implications), the error in the ratios was consistent across samples (*Figure 3D*) while the error in the proportions varied dramatically in both magnitude and direction (*Figure 3C*).

Our model explained almost all of the error in the measured compositions of the cell mixtures. We estimated bias from all samples by a simple point-estimation procedure (Materials and methods; *Table 1*). We then used the estimated bias to predict the observed compositions from the expected even mixtures using *Equation 2*. The measured pairwise ratios closely matched the ratios predicted by our model—the ratios of the efficiencies of the two taxa (black crosses in *Figure 3D*). The proportions predicted from the fitted model reduced the mean squared error by 98.8% and closely matched the observed proportions (*Figure 3B*).

The DNA and PCR-product mixture experiments confirmed that our model can also effectively describe partial MGS workflows. The compositions measured from DNA mixtures were affected by large systematic errors that were well explained by our model, while the systematic error in compositions measured from the PCR mixtures was small compared to the random errors (*Figure 3—figure supplement 1* and *Figure 3—figure supplement 2*). Notably, the bias in DNA mixtures substantially differed from the bias in the cell mixtures (*Table 1* and *Figure 3—figure supplement 1*). This observation suggests that PCR (performed in both experiments) and DNA extraction (performed only in the cell-mixture experiment) are both large, independent sources of bias that each act in accordance with our model.

Our model better explains the data from the cell and DNA mixtures than proposed alternatives while employing a small number of parameters (6, equal to the number of taxa minus 1). Two recent studies (*Krehenwinkel et al., 2017*; *Bell et al., 2019*) used simple linear regression of the observed proportion of a taxon against its actual proportion, which uses 7 or 14 parameters for all taxa depending on whether intercept terms are included. Such models do not constrain the observed proportions to the [0, 1] interval. More critically, they cannot explain that the observed proportion of a given taxon can be higher than or lower than its actual proportion in samples of different composition (e.g. *L. crispatus* in *Figure 3C*), while such behavior is a straightforward consequence of our model. *Brooks et al. (2015)* attempted to overcome this limitation by adding second and third-order interaction terms between taxa to the linear model. This model obtains a close fit at the cost of vastly increased model complexity—441 parameters for all taxa instead of just 6. As a result, the interactions model is likely to overfit and poorly predict the measured compositions with different compositions from which it is trained on. *Figure 3—figure supplement 3* compares the fit of our model to the simple linear model and the linear interactions model.

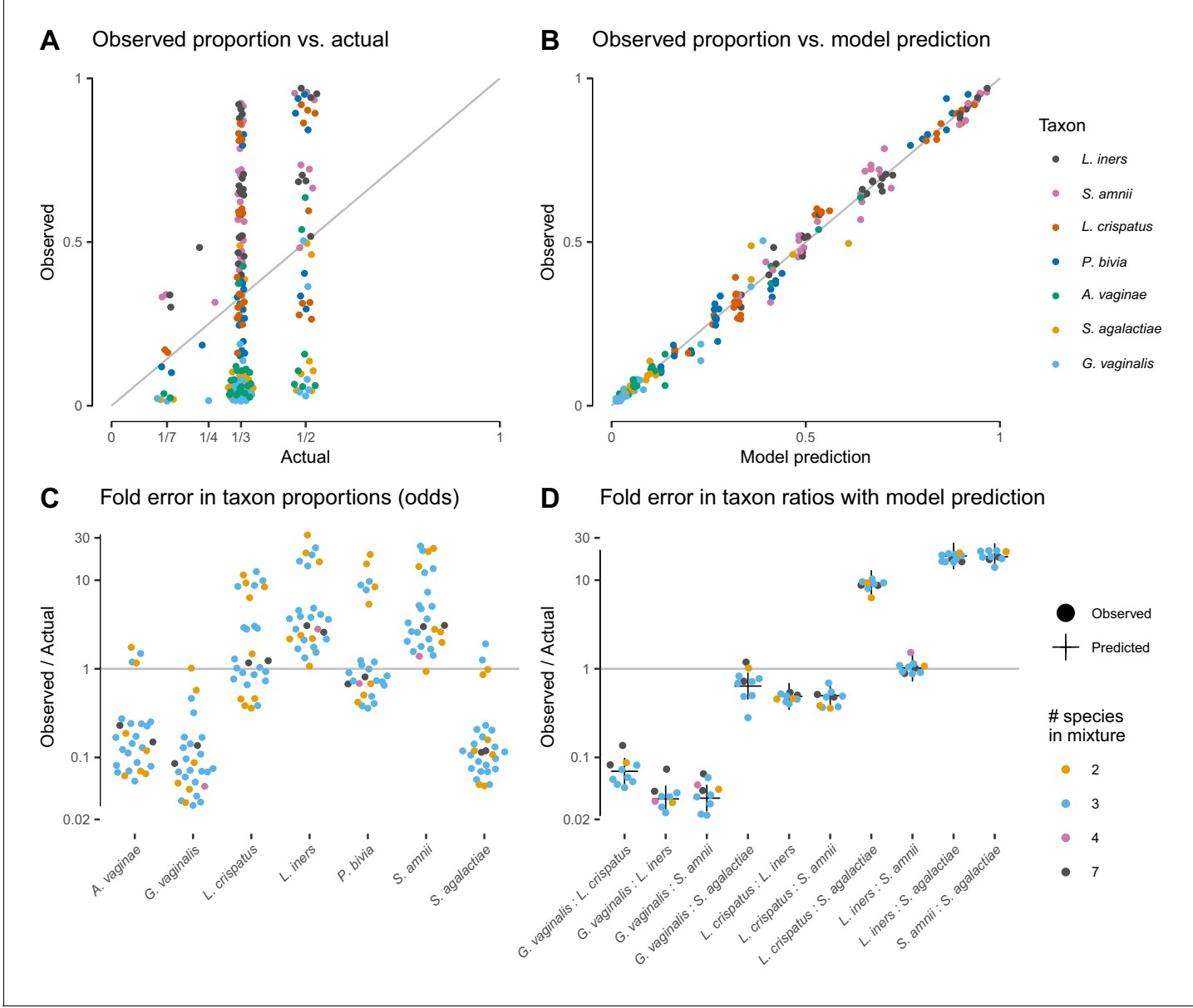

**Figure 3.** Our model of bias explains the systematic error observed in the *Brooks et al. (2015)* cell-mixture experiment. The top row compares the observed proportions of individual taxa to the actual proportions (Panel **A**) and to those predicted by our fitted bias model (Panel **B**). Panel **A** shows significant error across all taxa and mixture types that is almost entirely removed once bias is accounted for in Panel **B**. Panel **C** shows the observed error in proportions of individual taxa, while Panel **D** shows the error in the ratios of pairs of taxa for five of the seven taxa. The ratio predicted by the fitted model is given by the black cross in Panel **D**. As predicted by our model, the error in individual proportions (Panel **C**) depends highly on sample composition, while the error in ratios (Panel **D**) does not.

DOI: https://doi.org/10.7554/eLife.46923.004

The following figure supplements are available for figure 3:

**Figure supplement 1.** The observed error in taxon ratios for all three mixture experiments.

DOI: https://doi.org/10.7554/eLife.46923.005

**Figure supplement 2.** Observed vs. expected proportions under no bias, copy-number bias only, and the estimated bias.

DOI: https://doi.org/10.7554/eLife.46923.006

**Figure supplement 3.** Comparison between the simple linear model, the linear interactions model of *Brooks et al. (2015)*, and our model.

DOI: https://doi.org/10.7554/eLife.46923.007

**Table 1.** Estimated bias for the three *Brooks et al. (2015)* mixture experiments.
The first three columns show the bias estimated in each mixture experiment; the second three columns show the bias estimated for individual protocol steps from the mixture estimates. In each case, bias is shown as relative to the average taxon; that is, the efficiency of each taxon is divided by the geometric mean efficiency of all seven taxa. The last three rows summarize the multiplicative error in taxon ratios due to bias and noise. Taxa are ordered by decreasing efficiency in the cell mixtures. Abbreviations: PCR prod.: PCR product; Seq. + Inf.: Sequencing + Informatics.

| | Mixtures | | | Steps | | |
|---|---|---|---|---|---|---|
| Taxon | Cells | DNA | PCR prod. | Extraction | PCR | Seq.+Inf. |
| *Lactobacillus iners* | 4.7 | 2.3 | 1.2 | 2.0 | 1.9 | 1.2 |
| *Sneathia amnii* | 4.6 | 2.4 | 1.3 | 1.9 | 1.8 | 1.3 |
| *Lactobacillus crispatus* | 2.3 | 0.5 | 0.9 | 4.3 | 0.6 | 0.9 |
| *Prevotella bivia* | 1.8 | 0.4 | 0.9 | 4.6 | 0.4 | 0.9 |
| *Atopobium vaginae* | 0.3 | 1.1 | 1.0 | 0.3 | 1.0 | 1.0 |
| *Streptococcus agalactiae* | 0.2 | 2.0 | 0.9 | 0.1 | 2.2 | 0.9 |
| *Gardnerella vaginalis* | 0.2 | 0.4 | 0.8 | 0.4 | 0.5 | 0.8 |
| | | | | | | |
| Max pairwise bias | 29.3 | 6.1 | 1.6 | 36.6 | 5.2 | 1.6 |
| Avg. pairwise bias | 5.6 | 2.7 | 1.2 | 5.5 | 2.3 | 1.2 |
| Avg. pairwise noise | 1.2 | 1.2 | 1.3 | — | — | — |

DOI: https://doi.org/10.7554/eLife.46923.008

## Metagenomic sequencing of fecal samples with a spike-in mock community

The Phase III experiment of *Costea et al. (2017)* performed shotgun metagenomic sequencing of a cellular mock community spiked into fecal samples. The mock spike-in contained 10 bacterial taxa with known abundance spanning 2.5 orders of magnitude and (unintentionally) an *Escherichia/Shigella* contaminant with unknown true abundance. It was added to fecal specimens from eight individuals as well as a blank 'mock-only' sample. DNA was then extracted from each specimen using three distinct DNA extraction protocols (Protocols H, Q, and W) and measured via a common shotgun sequencing protocol. Here we test whether bias among the spike-in taxa is consistent across the varying backgrounds of the nine specimens.

Taxonomic profiles measured by MetaPhlAn2 (Materials and methods) showed substantial variation in the native bacterial composition across fecal specimens and in the proportion formed by the spike-in, both of which were protocol-dependent (*Figure 4A*). In contrast, the observed relative abundances of the spike-in taxa were consistent across specimens for a given protocol (*Figure 4A*). This observation is what we expect given that the true spike-in composition is fixed, since our model predicts that the error in the ratios among the spike-in taxa is independent of the presence and abundance of other taxa. *Figure 4A* shows abundances relative to the geometric mean of the 10 mock taxa. The average difference between the observed and actual abundance for each taxon estimates the bias of the protocol in terms of the taxon's efficiency relative to the average taxon (*Figure 4—figure supplement 1* and *Table 2*). The bias shows qualitative differences between protocols, with certain mock taxa being enriched by one protocol and diminished by another. Also, the consistent difference in the observed relative abundance of the contaminant indicates a consistent differential bias between Protocol W and the other two protocols of the contaminant relative to the 10 mock taxa. These results indicate a consistent and unique bias associated with each protocol when bias is measured in accordance to our model.

## Applications of the model

### Calibration

Our model implies that a protocol's bias can be estimated from control sample(s) of known composition and used to calibrate (through *Equation 9*) the measured compositions of unknown samples

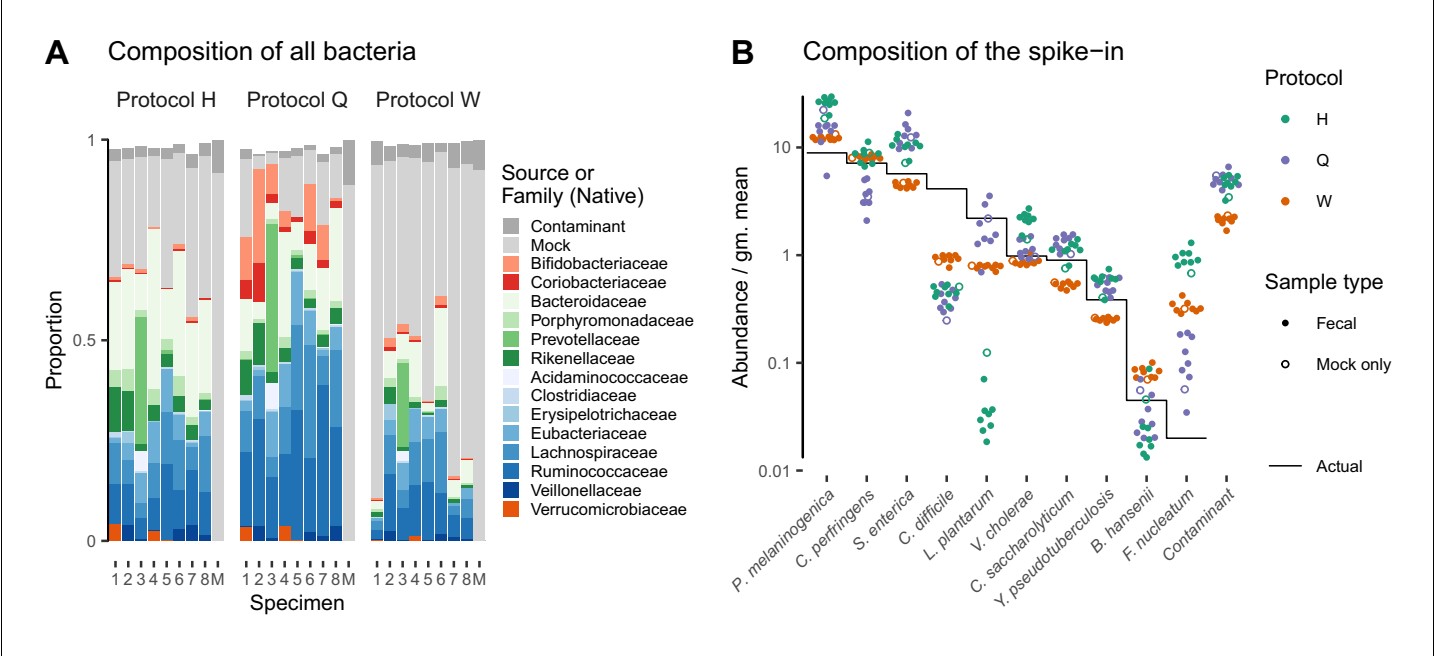

**Figure 4.** Bias of the mock spike-in in the *Costea et al. (2017)* experiment is consistent across samples with varying background compositions. Panel **A** shows the variation in bacterial composition across protocols and specimens (Labels 1 through 8 denote fecal specimens; M denotes the mock-only specimen) and Panel **B** shows the relative abundance of the 10 mock taxa and the spike-in contaminant (dots) against the actual composition (black line). In Panel **A**, color indicates source (mock, contaminant, or native gut taxon) and Family for native bacterial taxa with a proportion of 0.02 in at least one sample. Families are colored by phylum (Red: *Actinobacteria*, Green: *Bacteroidetes*, Blue: *Firmicutes*, Orange: *Verrucomicrobia*). In Panel **B**, abundance is divided by the geometric mean of the mock (non-contaminant) taxa in that sample.

DOI: https://doi.org/10.7554/eLife.46923.009

The following figure supplement is available for figure 4:

**Figure supplement 1.** Estimated bias for the mock taxa for the three protocols.
DOI: https://doi.org/10.7554/eLife.46923.010

towards their true compositions. In the *Brooks et al. (2015)* cell mixtures, we estimated the bias from two samples containing all seven taxa and used this estimate to calibrate the measurements of the other 69 samples. Calibration reduced the mean squared error of the proportions in the calibrated samples by 92.6% and the average Bray-Curtis dissimilarity between the actual and observed compositions from 0.35 to 0.08. In the *Costea et al. (2017)* dataset, the measured composition of the spike-in mock community deviated from the truth in a protocol-specific fashion (*Figure 5*, top row). We estimated the bias of each protocol on the mock taxa from three fecal specimens and used those estimates to calibrate all samples (Materials and methods). Calibration removed most of the systematic error and greatly increased the accuracy of the measurements (*Figure 5*, middle row).

Random error, or noise, in the measurement process creates error in the estimated bias that propagates into the calibrated measurements. To evaluate the effect of noise on the accuracy of bias estimation, we picked the protocol with an intermediate noise level (Protocol H) and estimated the standard error in the relative efficiencies as a function of the number of control samples (*Figure 5—figure supplement 1*). Because noise was much weaker than bias, standard errors were less than the bias even for a single control measurement, suggesting substantial benefits from calibration even with a limited number of control observations. The results further suggest that three or four control measurements for a taxon substantially reduces the risk of inaccurate bias estimates due to occasional large random errors.

Differential bias between experiments (*Equation 8*) can be estimated from samples common to each experiment, even if the actual composition of the common samples is unknown. Differential bias can then be used to calibrate measurements from various experiments to those of a chosen reference protocol, thereby making measured compositions from different experiments quantitatively

**Table 2.** Estimated bias and differential bias among the spike-in taxa for the three protocols (Protocols H, Q, and W) in the *Costea et al. (2017)* experiment.

The first three columns show the bias of the given protocol for the 10 mock taxa; the second three columns show the differential bias between protocols for the 10 mock taxa and the contaminant. In each case, bias is shown as relative to the average mock (non-contaminant) taxon; that is, the efficiency of each taxon is divided by the geometric mean efficiency of the 10 mock taxa. The last three rows summarize the multiplicative error in taxon ratios due to bias and noise; the contaminant is excluded from these statistics to allow direct comparison between bias and differential bias. Taxa are ordered as in *Figure 4B*.

| Taxon | Protocol | | | Protocol/Reference | | |
|---|---|---|---|---|---|---|
|  | H | Q | W | H/Q | H/W | Q/W |
| *Prevotella melaninogenica* | 2.81 | 1.55 | 1.37 | 1.82 | 2.05 | 1.12 |
| *Clostridium perfringens* | 1.18 | 0.49 | 1.14 | 2.41 | 1.04 | 0.43 |
| *Salmonella enterica* | 1.77 | 2.29 | 0.79 | 0.77 | 2.25 | 2.90 |
| *Clostridium difficile* | 0.11 | 0.09 | 0.22 | 1.24 | 0.49 | 0.40 |
| *Lactobacillus plantarum* | 0.02 | 0.77 | 0.35 | 0.02 | 0.05 | 2.18 |
| *Vibrio cholerae* | 2.10 | 1.16 | 0.89 | 1.81 | 2.37 | 1.31 |
| *Clostridium saccharolyticum* | 1.21 | 1.44 | 0.59 | 0.84 | 2.05 | 2.45 |
| *Yersinia pseudotuberculosis* | 1.46 | 1.35 | 0.66 | 1.08 | 2.21 | 2.05 |
| *Blautia hansenii* | 0.54 | 0.74 | 1.80 | 0.72 | 0.30 | 0.41 |
| *Fusobacterium nucleatum* | 46.29 | 4.98 | 16.51 | 9.30 | 2.80 | 0.30 |
| Contaminant | — | — | — | 0.89 | 2.13 | 2.38 |
|  |  |  |  |  |  |  |
| Max pairwise bias | 2751 | 56 | 74 | 428 | 59 | 10 |
| Avg. pairwise bias | 9.7 | 3.2 | 3.5 | 4.7 | 3.9 | 2.8 |
| Avg. pairwise noise | 1.3 | 1.5 | 1.1 | 1.5 | 1.3 | 1.5 |

DOI: https://doi.org/10.7554/eLife.46923.011

comparable even if their fidelity to the true compositions remains unclear. We illustrate calibration to a reference protocol using the multi-protocol design of the *Costea et al. (2017)* experiment (*Figure 5*). We defined the measurements by Protocol W as the reference composition that was then used in place of the actual composition in our calibration procedure. This greatly reduced the systematic differences between measurements from different protocols, without necessarily improving the accuracy compared to the actual composition (*Figure 5*, bottom row).

## Bias measurement as a means of evaluating and improving protocols

Under our model, the compositional vector of relative efficiencies completely describes the effect of bias in samples of any composition, and thus is the correct way to measure and evaluate bias.

For the purpose of selecting less-biased protocols, the overall magnitude of bias can be quantified through the use of ratio-based summary statistics. We provide two such statistics for the *Brooks et al. (2015)* experiment in *Table 1* and the *Costea et al. (2017)* experiment in *Table 2*. The maximum pairwise bias, equal to the geometric range of the relative efficiencies, indicates the maximum error due to bias in the ratio of any two taxa. The average pairwise bias indicates the magnitude of the multiplicative error averaged over all pairs of taxa. For the three shotgun protocols in *Table 2*, these statistics indicate that Protocol H has a much larger bias than the other two protocols, though one should keep in mind that the large values for Protocol H are heavily influenced by its extremely low efficiency for *L. plantarum* and high efficiency for *F. nucleatum*. Summarizing the residual error in the control samples leads to an analogous average pairwise measure of the noise, or random error, associated with each protocol, which we also include in the tables. The noise measure for the three shotgun protocols indicates that, in this case, the least biased protocol (Protocol Q) also yielded the noisiest measurements of the mock taxa. However, the average noise of the

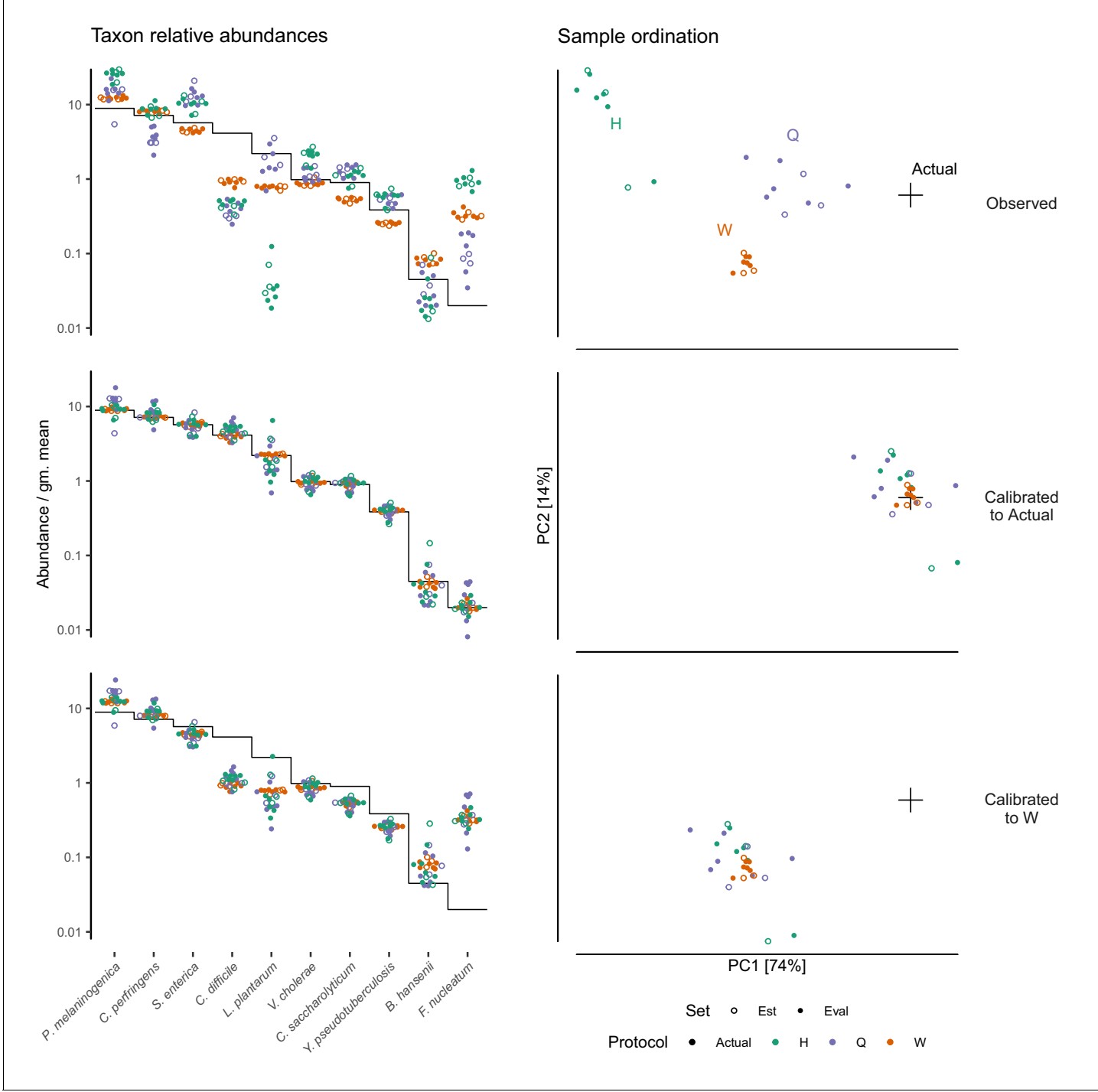

**Figure 5.** Calibration can remove bias and make MGS measurements from different protocols quantitatively comparable. For the sub-community defined by the mock spike-in of the *Costea et al. (2017)* dataset, we estimated bias from three specimens (the estimation set 'Est') and used the estimate to calibrate all specimens. The left column shows taxon relative abundances as in *Figure 4B* and the right column shows the first two principal components from a compositional principle-components analysis (*Gloor et al., 2017*). The top row shows the measurements before calibration; the middle, after calibration to the actual composition; and the bottom, after calibration to Protocol W.

DOI: https://doi.org/10.7554/eLife.46923.012

The following figure supplement is available for figure 5:

**Figure supplement 1.** Precision in the bias estimate vs. the number of control samples for Protocol H.

DOI: https://doi.org/10.7554/eLife.46923.013

protocols decreases with the propensity of the protocol to sequence the mock over the native gut taxa (*Figure 4A*), indicating that the greater noise of Protocol Q may be at least partially an artifact of limited sequencing depth.

Given suitable experimental designs, the estimated vectors of relative efficiencies can be used to quantify the bias attributable to specific parts of the workflow (Materials and methods and Appendix 1). In the *Brooks et al. (2015)* study, the same MGS workflow was run from different starting points: cells, extracted DNA, and PCR product. Comparing the bias resulting from different starting points leads to estimates of the bias attributable to DNA extraction, PCR amplification, and sequencing plus (bio)informatics (*Figure 3*; *Figure 6A*). For instance, dividing the relative efficiencies measured in the cell mixtures by those in the PCR mixtures provides an estimate of the bias that arises during DNA extraction. These estimates indicate that for these taxa and workflow, DNA extraction is the largest single source of bias, although PCR bias was also substantial. We can alternately understand these estimates through their predicted effect on the composition of an even mixture of taxa as it moves through the experimental workflow (*Figure 6B*), which clearly shows how extraction and PCR can oppose each other or work together. PCR and extraction bias acted in opposite directions for some taxa, such as *L. crispatus* and *P. bivia*, and in the same direction for others, such as *G. vaginalis* and for *L. iners*, leading to more moderate or extreme total relative efficiencies, respectively.

In the *Costea et al. (2017)* study, the same MGS workflow with different DNA extraction protocols was used to measure a common set of samples. This design implies that the differential bias between protocols (*Table 2*) can be attributed specifically to the effect of extraction (including possible effects of extraction on downstream steps; see Appendix 1). The differential bias of Protocol H relative to Protocols Q or W is substantially less than its bias (relative to the actual abundances), as can be seen from the summary statistics in *Table 2* and visually in *Figure 4—figure supplement 1*. This observation suggests that components of bias are shared between protocols, either due to similarities among the extraction protocols or bias from shared steps such as library preparation.

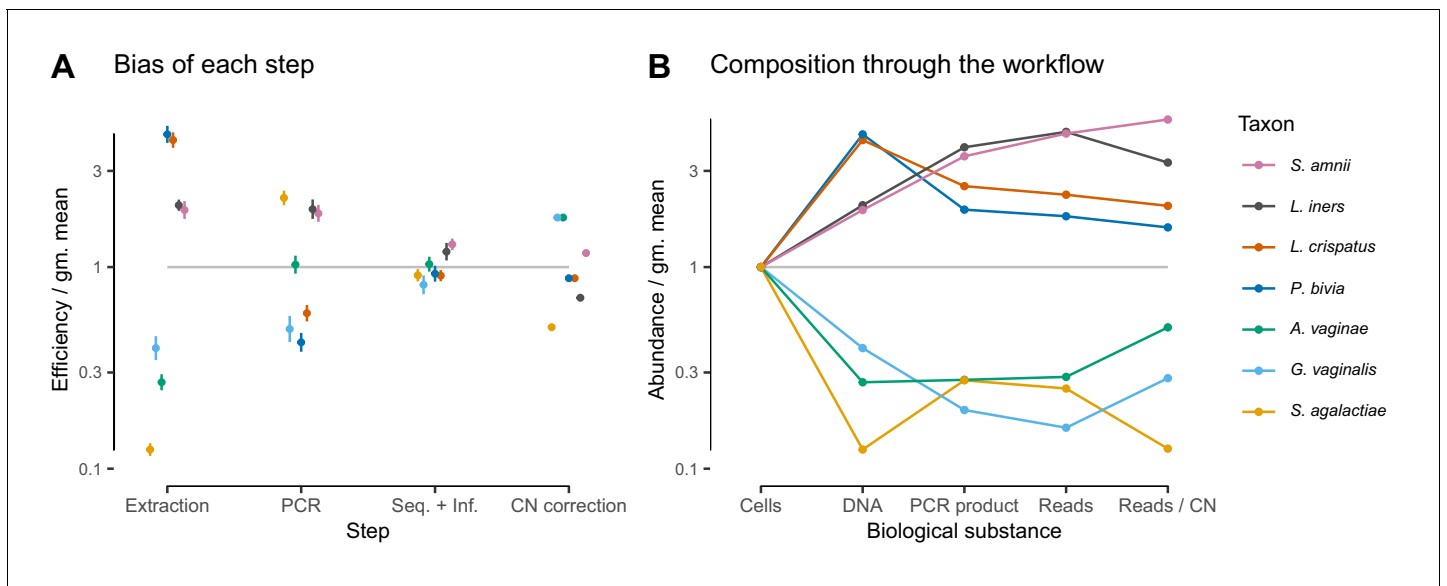

**Figure 6.** In the *Brooks et al. (2015)* experiment, bias is primarily driven by DNA extraction and is not substantially reduced by 16S copy-number (CN) correction. Panel **A** shows the bias estimate for each step in the experimental workflow (DNA extraction, PCR amplification, and sequencing + (bio) informatics), as well as the bias imposed by performing 16S CN correction (i.e. dividing by the estimated number of 16S copies per genome). Bias is shown as relative to the average taxon—that is, the efficiency of each taxon is divided by the geometric mean efficiency of all seven taxa—and the estimated efficiencies are shown as the best estimate multiplied and divided by two geometric standard errors. Panel **B** shows the composition through the workflow, starting from an even mixture of all seven taxa, obtained by sequentially multiplying the best estimates in Panel **A**.
DOI: https://doi.org/10.7554/eLife.46923.014

The following figure supplement is available for figure 6:

**Figure supplement 1.** PCR bias and total bias vs. bias predicted by 16S copy number.
DOI: https://doi.org/10.7554/eLife.46923.015

Estimates of bias can be used to test mechanistic hypotheses and proposed methods for predicting bias. To demonstrate this application, we considered the effect of 16S copy number (CN) on bias in the *Brooks et al. (2015)* data. We estimated 16S CN per genome and per bp for the seven taxa using available CN and genome size estimates (Materials and methods; *Table 3*). We then compared the bias predicted by CN to the estimated PCR bias and the estimated bias of the total protocol *without* CN correction (*Figure 6—figure supplement 1*). 60% of the variance in estimated PCR bias was explained by CN variation (log relative efficiency scale; coefficient of determination $\approx 0.60$; $p \approx 0.021$ by permutation test). In contrast, total bias was poorly explained by CN variation (coefficient of determination $\approx 0.10$; $p \approx 0.23$ by permutation test). Accordingly, CN correction reduced the mean squared error in the taxon proportions by about half in the DNA mixtures but only slightly in the cell mixtures (*Figure 3—figure supplement 2*). The limited effect of CN correction can also be seen in *Figure 6B*. These results indicate that CN variation is just one component of PCR bias, which itself is just one component of total bias, and thus even perfect correction of CN bias may not substantially ameliorate total bias in marker-gene sequencing experiments.

## Discussion

The lack of a rigorous understanding of how bias distorts marker-gene and metagenomic sequencing (jointly, MGS) measurements stands in the way of accurate and reproducible community-composition measurements. Previous analyses of bias in MGS experiments have largely relied on descriptive statistical models (*Brooks et al., 2015*; *Sinha et al., 2017*; *Krehenwinkel et al., 2017*; *Bell et al., 2019*) whose parameters cannot be identified with biophysical quantities that one might expect to apply to differently composed samples. Failure to develop more mechanistic models may have stemmed from the seeming hopelessness of accounting for the many verified sources of bias. Here we proposed a mathematical model of bias in MGS experiments as a set of taxon-specific factors (the relative efficiencies) that multiply the true relative abundances to produce the measured relative abundances. Our model was inspired by the observation that many sources of bias, such as differences in DNA extraction efficiency (*Morgan et al., 2010*), PCR primer binding and amplification efficiency (*Wagner et al., 1994*; *Suzuki and Giovannoni, 1996*; *Polz and Cavanaugh, 1998*; *Edgar, 2017*), and marker-gene copy number (*Kembel et al., 2012*), are thought to act multiplicatively, and hence so could their cumulative effect. The parameters in our model (the relative efficiencies) have biophysical interpretations as the relative yield per unit of input for each taxon, for individual steps or for the workflow overall. Our hypothesis that the relative efficiencies are consistent across samples is grounded in existing understanding of individual bias mechanisms and was supported by marker-gene and shotgun-metagenomic sequencing measurements of mock bacterial communities with varying composition. We further showed how our model could be used to measure, understand, and correct bias.

We found bias to be independent of sample composition only after accounting for the compositional nature of MGS measurements. Bias appeared inconsistent when viewed in terms of taxon proportions—for example the measured proportion of *L. crispatus* was both higher and lower than its

**Table 3.** Estimated genome size and 16S copy number for the seven mock taxa in the *Brooks et al. (2015)* experiment (Materials and methods).

| Taxon | Genome size (Mbp) | Copy number |
|---|---|---|
| *Atopobium vaginae* | 1.44 | 2* |
| *Gardnerella vaginalis* | 1.64 | 2 |
| *Lactobacillus crispatus* | 2.04 | 4 |
| *Lactobacillus iners* | 1.28 | 5* |
| *Prevotella bivia* | 2.52 | 4* |
| *Sneathia amnii* | 1.33 | 3* |
| *Streptococcus agalactiae* | 2.16 | 7 |

*Denotes copy numbers that were instead estimated to be 1 by *Brooks et al. (2015)*.
DOI: https://doi.org/10.7554/eLife.46923.016

true value in different samples (*Figure 3C*). However, these apparent inconsistencies did not reflect inconsistency in the action of bias, but instead were a consequence of the compositional nature of MGS data. A limited number of sequencing reads are generated from each sample, so if one taxon is enriched by bias then other taxa must be correspondingly diminished. Therefore, a taxon's proportional over- or under-representation depends not on its absolute measurement efficiency but on its efficiency relative to the average individual (e.g. microbial cell) in the sample—*L. crispatus* increased in proportion when its efficiency was greater than the sample average, and decreased otherwise. Models that do not account for this effect (such as those of *Brooks et al., 2015*; *Krehenwinkel et al., 2017*; *Bell et al., 2019*; *Kevorkian et al., 2018*) will yield parameter estimates that do not extrapolate to differently-composed samples. Once we accounted for compositionality it became clear that relative efficiencies were consistent across differently composed samples. Bias had the same effect in each sample when we divided out the effect of compositionality by considering ratios of taxa (*Figure 3D*), and when we fully modeled the normalization involved in constructing proportions we found that a single set of relative efficiencies explained the observed proportions in every sample (*Figure 3B*).

A quantitative model allows the sensitivity of downstream analyses to bias to be rigorously evaluated. Consider the often unstated assumption that analyses of the differences between samples measured in the same experiment should be robust to bias, because each sample is biased in the same way. We can formally evaluate this assumption in the simple numerical example shown in *Figure 2*: Sample $S_1$ has higher Shannon diversity than Sample $S_2$ but the measured diversity of Sample $S_1$ is lower than Sample $S_2$, and Sample $S_1$ has a higher proportion of Taxon 3 than Sample $S_2$ but is observed to have a lower proportion of Taxon 3, despite the same bias distorting each sample. Whether such qualitative errors are likely can be investigated by simulating our model with empirical distributions of bias and community compositions. For example, consider the *Bacteroidetes:Firmicutes* ratio, a repeatedly-proposed diagnostic of gut health (*Ley et al., 2006*; *Finucane et al., 2014*). The range of relative efficiencies within the *Firmicutes* indicates that the *Bacteroidetes:Firmicutes* ratio measured by the metagenomic sequencing protocols evaluated in *Costea et al. (2017)* can differ from the true ratio by very little, or by as much as 100-fold, depending on which Firmicutes species is dominant in the sample!

If bias acts (as we propose) as a consistent multiplication of the relative abundances, then analyses of MGS data based on taxon ratios could reduce the possibility for spurious results and make results from different experiments more comparable. The key insight is that the fold changes between samples in the ratios between taxa is invariant to consistent multiplicative bias, because the relative efficiencies divide out (*Equation 6* and *Equation 7*). In contrast, such canceling does not occur for the fold-changes in taxon proportions, as exemplified by the spurious observed increase of Taxon 2 in *Figure 2*. To be clear, this bias-invariance property for ratio-based analyses only holds for samples biased in the same way, so these analyses still must be conducted within experiments sharing a common MGS protocol. But by controlling for study-specific bias, these analyses may give results that are more concordant across studies than other analyses. One ready source of such methods is the field of Compositional Data Analysis (CoDA) (*Aitchison, 1986*; *Gloor et al., 2017*). In fact, our model of bias is equivalent to what is referred to as a compositional perturbation in the CoDA field; many CoDA methods are invariant to compositional perturbations (*Aitchison, 2003*; *van den Boogaart and Tolosana-Delgado, 2013*) and thus would be invariant to bias. CoDA methods are being increasingly used to analyze MGS data, but to date this has been motivated by the need to account for the compositionality of MGS data. The possibility that such methods could also reduce or remove the effect of bias has not been widely appreciated.

Studies investigating bias and/or optimizing protocols should evaluate the systematic errors introduced by bias in a way that accounts for the compositional nature of MGS data. Most previous studies of bias quantified bias with taxon proportions (e.g. *Brooks et al., 2015*; *Krehenwinkel et al., 2017*; *Bell et al., 2019*) or proportion-based summary statistics such as Bray-Curtis dissimilarities (*Sinha et al., 2017*) or differences in Shannon diversity (*Song et al., 2016*). Proportion-based measurements do not consistently measure bias in differently composed samples, and thus are difficult to interpret and can mislead researchers attempting to reduce the effect of bias on their experiments. The adoption of compositionally aware analytical methods to study bias may lead to insights that generalize beyond the specific sample compositions considered in these studies. In particular, quantification of bias in the form of the 'bias vector' of relative efficiencies we proposed here has a

natural biological interpretation as the relative yields of each taxon and can be naturally decomposed into an element-wise product of bias vectors for each step in a workflow, allowing for granular investigation of MGS protocol choices.

Our results suggest that calibration could become a practical approach to improving the accuracy of MGS analyses and diagnostics. If bias is composition independent, then it can be estimated from one or more control samples and those estimates used to correct the relative abundances in target samples. Our results overcomes major limitations in recent attempts at MGS calibration (*Brooks et al., 2015*; *Thomas et al., 2016*; *Krehenwinkel et al., 2017*; *Bell et al., 2019*) by indicating how to obtain a compositionally independent estimate of bias for many taxa from a small number of control samples. Intriguingly, we show that the differential bias between protocols behaves in the same manner as the bias of an individual protocol. This property opens the possibility of calibration based on a reference protocol's measurements of control samples even if their true composition is not known. Reference calibration does not give the abundances in terms of biologically tangible units like cell concentration, but can make measurements from differently biased experiments quantitatively comparable, allowing diagnostic criteria to be applied outside of the lab in which they were defined. Reference calibration may sidestep the practical challenges of creating defined cellular mixtures of many taxa by using natural samples (or aggregations of natural samples) as calibration controls that would then contain the full range of taxa naturally present.

## Limitations and next steps

We found bias to act multiplicatively in accordance with our model in two mock-community experiments; however, many sources of bias may deviate from multiplicativity under non-ideal conditions. For example, it has been observed that the efficiency of a target sequence is altered by saturation during PCR amplification when the target is either rare or highly abundant (*Suzuki and Giovannoni, 1996*; *Gonzalez et al., 2012*). In a non-microbial marker-gene experiment, *Thomas et al. (2016)* found a strong and consistent saturation effect that may have been caused by such a mechanism. The opposite of saturation, where taxa have lower efficiencies when rarer in the sample, may occur if low-abundance taxa are culled by the minimum-abundance thresholds used by taxonomic profilers such as MetaPhlAn2 (*Truong et al., 2015*). Such deviations from multiplicativity may be eliminated through protocol design (e.g. to avoid PCR saturation; *Polz and Cavanaugh, 1998*) or accounted for with extensions allowing for deterministic and random variation in efficiencies.

Even when bias does act multiplicatively on individual microbial taxa, that multiplicativity will not hold for aggregates of taxa that vary in their efficiencies (Appendix 1). Consider again the *Bacteroidetes:Firmicutes* ratio diagnostic. We know that within the *Firmicutes* phylum there is tremendous phenotypic variation, and that variation manifested itself in the *Costea et al. (2017)* study as order-of-magnitude differences between the relative efficiencies of various *Firmicutes* species. As a result, consistent multiplicative bias could dramatically enrich or diminish the relative abundance of the *Firmicutes* phylum depending on which *Firmicutes* species were present. Unfortunately, the potential for bias to act inconsistently on aggregates of taxa cannot be entirely circumvented by eschewing aggregation in our analyses because even the fundamental units we derive from MGS data effectively aggregate variation at some level (*McLaren and Callahan, 2018*). In the bacterial context, the traditional 97% ribosomal OTU groups variation at roughly the genus level (*Yarza et al., 2014*), exact short-read ribosomal sequence variants and common shotgun profilers group variation at roughly the species level (*Edgar, 2018*; *Hillmann et al., 2018*), and metagenome assembly combines strains too similar to be separated by the assembler or by the subsequent binning (*Dick, 2018*, p. 79). This does not make bias an intractable problem, but it does mean that additional work is needed to understand the phylogenetic scales over which bias significantly varies. Methods to control bias will need to operate on taxonomic objects with commensurate or finer resolution than that of variation in the bias phenotype (*McLaren and Callahan, 2018*).

Cellular phenotype and matrix chemistry can vary substantially between samples of different types, which may cause bias to vary among samples. For example, different cellular growth states may change the cellular membrane and thus the efficiency of DNA extraction for a given taxon, and inhibitors in soil samples are known to influence the efficiency of PCR (*Schrader et al., 2012*). In the *Costea et al. (2017)* experiment, protocols differentially extracted the mock and native gut taxa (*Figure 4A*), which may have been due to physiological differences between the lab-grown cells and the preserved fecal cells. Potential phenotype and matrix effects are particularly relevant for

calibration applications, as accurate calibration will require that bias measured in the controls is representative of the bias in the target samples.

The development of effective calibration methodologies will be limited by our ability to develop control samples that cover the range of taxa present in target communities. The bias estimation procedure we developed here is limited to those taxa present in the controls, and thus only allows for partial calibration of the subcomposition of the target samples consisting of those taxa. However, even partial calibration is useful when controls with key taxa in the target community are available. For example, vaginal microbiome samples from a patient over several clinical visits analyzed by *Brooks et al. (2015)* were mostly comprised of the seven taxa in their mock mixtures. It may be possible to effectively augment the range of taxa with credible bias estimates beyond those included in control samples by using phylogenetic inference methods to predict the bias of related taxa (*Goberna and Verdú, 2016*). It will be easier to broadly cover the taxa in a given environment when calibrating to a reference protocol, because any sample measured by the reference protocol can be used as a reference calibration control, including samples from the environment of interest.

Our discussion has so far ignored another ubiquitous source of error in MGS experiments—contamination, either from the lab, reagents, or other samples (*Eisenhofer et al., 2019*). Contamination may be more important than bias in certain scenarios, such as the study of low biomass samples or analyses that are sensitive to very low frequency taxa, and bias correction may be insufficient for accurate measurement and inference in such settings. Accurate bias inference also requires accurate removal of contaminant reads prior to estimation through manual filtering or automated removal methods (*Davis et al., 2018*; *Larsson et al., 2018*). Incorporation of our bias model with models of contamination may allow for simultaneous and improved estimation and correction of both error processes.

The model of bias we explored in this paper treats bias and measurement error more generally as deterministic. In practice, however, there is variability in the error of taxon ratios around the mean (*Figure 3D*; *Figure 4—figure supplement 1*). We developed a point-estimation method with a bootstrapping procedure to estimate the bias with associated uncertainty from random observations (Materials and methods). However, robust estimation and calibration may require a statistical model of bias that explicitly accounts for sources of random error such as variation in relative efficiencies across samples and limited sequencing depth. A statistical model would facilitate the construction of confidence intervals for the calibrated taxon proportions in a sample. Challenges associated with building such a model include modeling the presence of taxa thought to be absent from the community (but observed due to contamination or index switching; *Eisenhofer et al., 2019*), the absence of taxa known to be present (*Yeh et al., 2018*), and accounting for the noise associated with the count nature of sequencing data. Our finding that multiplicative error in taxon ratios provides a parsimonious model for bias paves the way for the development of a such a statistical model, which we leave for future work.

## Conclusion

We suggest a simple yet profound change in how researchers view MGS measurements. Currently, researchers tend to either 1) take MGS measurements as telling us only about presence and absence of detectable taxa, 2) hope that bias in the measurements of individual samples will somehow cancel out when analyzing differences between samples within a given experiment, or 3) pretend bias doesn't exist. We propose a new view in which the measured relative abundances within an experiment are biased by unknown—but constant—multiplicative factors. When bias acts consistently in this manner it can be accounted for through the use of bias-insensitive analyses or corrected by a calibration procedure. Our results lay a foundation for the rigorous understanding of bias in marker-gene and metagenomic sequencing measurements that is required for accurate and reproducible research using MGS methods and for the development of reliable MGS diagnostics and interventions.

## Materials and methods

### Bias estimation

We describe a procedure for averaging multiple control observations to obtain a single estimate of the protocol's bias; additional details and motivation are given in Appendix 2. For sample $s$ in a set of control samples $S$, let $\mathbf{A}(s)$ denote the actual composition and $\mathbf{O}(s)$ be the observed composition. We assume that $\mathbf{O}(s)$ and $\mathbf{A}(s)$ are non-zero for the same taxa; in practice, this assumption requires ignoring sequencing reads from taxa not supposed to be in the sample and adding a small abundance to taxa actually present but not detected. Under our deterministic model, the observed composition is given exactly by *Equation 2*. In practice, however, each measurement will vary—for example, due to random error in sample construction, variation in sample handling, and random sampling of reads during sequencing. We decompose the error $\mathbf{O}(s)/\mathbf{A}(s)$ into a deterministic component $\mathbf{B}$ and a random component $\epsilon(s)$,

$$\mathbf{O}(s) \sim \mathbf{A}(s) \cdot \mathbf{B} \cdot \epsilon(s). \tag{10}$$

The random error $\epsilon(s)$ is a random compositional vector that we assume has an expected value of $(1,\ldots,1)$ in the compositional Aitchison geometry (given by the element-wise geometric mean; *Aitchison, 2003*, p. 38). Intuitively, we estimate $\mathbf{B}$ by the vector $\hat{\mathbf{B}}$ that minimizes the residual errors, $\hat{\epsilon}(s) \sim \mathbf{O}(s)/(\mathbf{A}(s) \cdot \hat{\mathbf{B}})$.

We quantify the magnitude of errors by the *Aitchison norm*. The Aitchison norm (*Pawlowsky-Glahn, 2015*; *Aitchison, 2003*, Chapter 2) of a $K$-element composition $\mathbf{X}$ is given by

$$\|\mathbf{X}\| = \sqrt{\frac{1}{K} \sum_{i<j} \left[\ln \frac{X_i}{X_j}\right]^2} = \sqrt{\sum_{i=1}^{K} \left[\ln \frac{X_i}{g(\mathbf{X})}\right]^2}, \tag{11}$$

where $g(\mathbf{X}) = \left(\prod_{i=1}^{K} X_i\right)^{1/K}$ is the geometric mean of the elements of $\mathbf{X}$. When estimating the bias from a sample $s$ that is missing some of the taxa, the elements of $\mathbf{O}(s)/\mathbf{A}(s)$ and of the residual error $\hat{\epsilon}(s)$ corresponding to the missing taxa are undefined. We define $\|\hat{\epsilon}(s)\|$ in this case by restricting to just the defined elements (and adjusting $K$ accordingly) before applying *Equation 11*.

We take our estimate of $\mathbf{B}$ to be the compositional vector that minimises the sum of squared residual error over all samples,

$$\hat{\mathbf{B}} \sim \underset{\mathbf{B}}{\arg\min} \sum_{s \in S} \|\mathbf{O}(s)/(\mathbf{A}(s) \cdot \mathbf{B})\|^2. \tag{12}$$

This definition equates $\hat{\mathbf{B}}$ with the compositional mean, or *center*, of the compositional errors $\mathbf{O}(s)/\mathbf{A}(s)$ when the center is defined to allow missing values (Appendix 2). If all samples contain all $K$ taxa, then $\hat{\mathbf{B}}$ is given by the element-wise geometric mean of the set $\{\mathbf{O}(s)/\mathbf{A}(s)\}$,

$$\hat{\mathbf{B}} \sim \left( \left[\prod_{s \in S} \frac{O_1(s)}{A_1(s)}\right]^{1/|S|}, \ldots, \left[\prod_{s \in S} \frac{O_K(s)}{A_K(s)}\right]^{1/|S|} \right), \tag{13}$$

where $|S|$ is the number of samples in the set $S$. More generally, the solution to $\hat{\mathbf{B}}$ can be computed using the projection approach of *van den Boogaart et al. (2006)*. The solution is unique up to compositional equivalence given sufficient taxonomic overlap among the samples (Appendix 2).

Differential bias of the given protocol to a reference Protocol $R$ that has also measured the samples in $S$ can be estimated without knowing the actual sample compositions by replacing the actual compositions $A(s)$ in the above with the reference measurements $\mathbf{O}^{(R)}(s)$.

We describe a bootstrap procedure for estimating the uncertainty of the bias estimate in Appendix 2. Each bootstrap replicate consists of drawing either multinomial or Dirichlet weights for the control samples and computing the weighted center to obtain the replicate value of $\hat{\mathbf{B}}$. Standard errors for the relative efficiencies are estimated by the geometric standard deviation of the corresponding efficiency ratio across replicates.

## Bioinformatics and statistical analysis

Data and code availability

Functions and a tutorial for estimating and visualizing bias and performing calibration are provided in the 'metacal' R package, available on GitHub (*McLaren, 2019a*, copy archived at https://github.com/elifesciences-publications/metacal). The raw data for our analysis is available in the 'Additional files' of *Brooks et al. (2015)* and in European Nucleotide Archive study accession PRJEB14847 for *Costea et al. (2017)*. The processed data—along with all code used to download and process the raw data, perform all statistical analyses, and generate all figures and tables—is contained in the manuscript's GitHub repository (*McLaren, 2019b*, copy archived at https://github.com/elifesciences-publications/mgs-bias-manuscript). Analysis and visualization is performed using the R software environment (*R Development Core Team, 2018*) with the 'metacal' package, the 'tidyverse' suite of R packages (*Wickham, 2019*), and the 'cowplot' R package (*Wilke, 2019*). Analysis code is contained in R-markdown documents that can be executed to generate all numerical results, tables, and figures. Versions that have been 'knit' into html documents showing code interlaced with output and figures are available in the GitHub repository.

### *Brooks et al. (2015)* experiment

We used taxonomic profiles generated in the original study and provided as supplemental information. Specifically, we used the sample information and read assignments in Additional Files 2, 10, and 11 of *Brooks et al. (2015)* to build a table of amplicon sequences assigned to each of the seven mock taxa in each sample. *Brooks et al. (2015)* used a classification method and 16S reference database designed for species-level classification of vaginally associated taxa from V1-V3 region amplicons (*Fettweis et al., 2012*). Reads were assigned to species in the database according to a 97% sequence identity threshold, resulting in 93.5% of reads assigned and for which the vast majority (99.98%) were assigned to species corresponding to the seven mock taxa. We discarded the small fraction (0.0002%) of reads assigned to other species. Most samples were assigned a small fraction of their reads from species not expected to be in the sample. These out-of-sample species generally had much lower frequency than the expected species, suggesting they were the result of cross-sample contamination rather than mislabeling or misconstruction of the samples. We therefore removed these reads before evaluating and estimating bias.

We took the actual composition of each sample to be an even mixture of the taxa added to that sample, in units of cell concentration, DNA concentration, or PCR-product concentration. *Brooks et al. (2015)* constructed the cell mixtures to be even mixtures based on CFUs (a proxy for cell concentration); the DNA mixtures based on DNA concentration; and the PCR mixtures based on volume from amplification of a fixed weight of DNA. Extraction and PCR protocols differed somewhat when using pure cultures to create the DNA and PCR-product mixtures than when applied to communities in the cell experiment. Thus, the DNA and PCR product in the second and third experiments may differ qualitatively from that in the cell mixture experiments, which could in principle affect the bias of downstream steps.

We estimated genome size and 16S copy number for the seven mock taxa from available genome databases and experimental measurements. We estimated genome size by the average genome size for the given species in NCBI RefSeq release 86 as collated by the GTDB (*Parks et al., 2018*). We estimated 16S copy number (CN) through a combination of RefSeq annotations for the given species; CN estimates in the rrnDB for the given species or their nearby relatives identified in the GTDB phylogeny (*Parks et al., 2018*); and measurement by pulse-field gel electrophoresis reported by *Yuan et al. (2012)* for *A. vaginae* and *L. iners*. A full account is given in the manuscript GitHub repository. The resulting genome size estimates approximately agree with those of *Brooks et al. (2015)*, but the 16S CN estimates differ substantially for several taxa (compare our *Table 3* to their Table 5). In particular, *Brooks et al. (2015)* estimated four taxa to have CNs of 1 based on NCBI annotations then available, but we suspect these numbers to be artifacts of poor assembly and annotation.

We estimated the bias predicted due to CN variation for each mixture type as follows. Cell mixtures: CN bias is simply the compositional vector of CNs (16S copies per genome). DNA mixtures: CN bias is the compositional vector of CN divided by genome size (16S copies per bp). PCR-product

mixtures: CN bias is the compositional identity vector (1,..., 1) (i.e. no bias). Denoting the estimated CN bias as $\hat{\mathbf{B}}^{(\text{CN})}$ for the given experiment, the CN-corrected proportions are $Pr[\mathbf{O}/\hat{\mathbf{B}}^{(\text{CN})}]$.

For each mixture experiment, we estimated bias as described above in 'Bias estimation'. We then used these estimates to partition our estimate of the total protocol bias into three steps—1) DNA extraction, 2) PCR amplification, and 3) sequencing and bioinformatics—under the simplifying assumption that the bias of shared steps are the same across experiments. We assume the bias of the cell mixture experiments is $\mathbf{B}^{(\text{Cells})} = \mathbf{B}^{(P_1)} \cdot \mathbf{B}^{(P_2)} \cdot \mathbf{B}^{(P_3)}$, of the DNA mixtures is $\mathbf{B}^{(\text{DNA})} = \mathbf{B}^{(P_2)} \cdot \mathbf{B}^{(P_3)}$, and of the PCR-product mixtures is $\mathbf{B}^{(\text{PCR product})} = \mathbf{B}^{(P_3)}$. We therefore estimate the extraction bias as $\hat{\mathbf{B}}^{(P_1)} = \hat{\mathbf{B}}^{(\text{Cells})}/\hat{\mathbf{B}}^{(\text{DNA})}$, the PCR bias as $\hat{\mathbf{B}}^{(P_2)} = \hat{\mathbf{B}}^{(\text{PCR product})}/\hat{\mathbf{B}}^{(\text{DNA})}$, and the sequencing and bioinformatics bias as $\hat{\mathbf{B}}^{(P_3)} = \hat{\mathbf{B}}^{(\text{PCR product})}$.

### *Costea et al. (2017)* Phase III experiment

We downloaded raw sequencing reads for the Phase III experiment from European Nucleotide Archive study accession PRJEB14847 and generated taxonomic profiles using MetaPhlAn2 version 2.7.6 (*Truong et al., 2015*) with the command-line options `–min_cu_len 0 –stat avg_g`. These options were chosen to increase sensitivity and accuracy for the rarest spike-in taxa and resulted in the detection of all spike-in taxa in every sample. Taxonomic profiles generated by MetaPhlAn2 provide estimated proportions of taxa at various taxonomic levels. We restricted our analysis to species-level abundances and the kingdom Bacteria, which constituted over 99% of non-viral abundance in each sample.

*Costea et al. (2017)* reported *Escherichia coli* as a likely spike-in contaminant due to its presence in sequence data from the mock-only samples. Consistent with this report, the MetaPhlAn2 profiles showed a substantial presence of *Shigella flexneri* in the mock-only samples and we identified this species as the 'Contaminant' in our subsequent analyses and in all figures and tables.

We estimated the true mock-community composition using the flow cytometry (FACS) measurements reported in *Costea et al. (2017)*. We used the arithmetic mean of two replicate measurements where available and ignored any measurement error in the resulting actual mock composition for our analysis. The FACS measurements provided by *Costea et al. (2017)* disagree with those shown in their Figure 6 for three taxa (*V. cholerae*, *C. saccharolyticum*, and *Y. pseudotuberculosis*). Analysis of our MetaPhlAn2 profiles indicates that these taxa are most likely mislabeled in the figure and not in the FACS measurements. A mislabeling in the FACS measurements would change the specific bias values we estimate for these taxa but not our main results or conclusions.

We estimated the bias of each protocol and the differential bias between protocols as described in 'Bias estimation'. We estimated standard errors using the Dirichlet-weighted bootstrap method described in Appendix 2. To determine how precision in the bias estimate for Protocol H varies with the number of control samples (*Figure 4—figure supplement 1*), we computed standard errors using the multinomial-weighted bootstrap method with the number of trials in the multinomial distribution equal to the specified number of control samples (Appendix 2).

To demonstrate calibration, we randomly chose three fecal specimens to use as the 'estimation set' to estimate bias, and then calibrated all samples using *Equation 9*. We excluded the mock-only specimen from the estimation set since its atypical values for a few taxa resulted in an unrepresentative picture of the success of calibration; however, we included it when evaluating the effect of noise on bias estimation in *Figure 4—figure supplement 1*.

## Acknowledgements
We thank Glen Satten and David Clausen for valuable discussions.

# Additional information

## Funding
The authors declare that there was no funding for this work.

## Author contributions
Michael R McLaren, Data curation, Formal analysis, Validation, Investigation, Visualization, Methodology, Writing—original draft, Writing—review and editing; Amy D Willis, Methodology, Writing—original draft, Writing—review and editing; Benjamin J Callahan, Conceptualization, Supervision, Funding acquisition, Investigation, Methodology, Writing—original draft, Writing—review and editing

## Author ORCIDs
Michael R McLaren (iD) https://orcid.org/0000-0003-1575-473X
Amy D Willis (iD) https://orcid.org/0000-0002-2802-4317
Benjamin J Callahan (iD) https://orcid.org/0000-0002-8752-117X

## Decision letter and Author response
Decision letter https://doi.org/10.7554/eLife.46923.026
Author response https://doi.org/10.7554/eLife.46923.027

# Additional files

## Supplementary files
• Transparent reporting form
DOI: https://doi.org/10.7554/eLife.46923.017

## Data availability
All data analysed in this study are publicly available through the 'Additional files' of http://www.bio-medcentral.com/1471-2180/15/66 (data derived from NCBI BioProject PRJNA267701) and through ENA Study PRJEB14847.

The following previously published datasets were used:

| Author(s) | Year | Dataset title | Dataset URL | Database and Identifier |
|---|---|---|---|---|
| Brooks JP, Edwards DJ, Harwich MD, Rivera MC, Fettweis JM, Serrano MG, Reris RA, Sheth NU, Huang B, Girerd P, Strauss JF, Jefferson KK, Buck GA | 2015 | Quantifying Bias in 16S rRNA Experiments due to DNA Extraction, PCR Amplification, and Sequencing and Classification | https://www.ncbi.nlm.nih.gov/bioproject/267701 | NCBI BioProject, PRJNA267701 |
| Costea PI, Zeller G, Sunagawa S, Pelletier E, Alberti A, Levenez F, Tramontano M, Driessen M, Hercog R, Jung F-E, Kultima JR, Hayward MR, Coelho LP, Allen-Vercoe E, Bertrand L, Blaut M, Brown JRM, Carton T, Cools-Portier S, Daigneault M, Derrien M, Druesne A, de Vos WM, Finlay BB, Flint HJ, Guarner F, Hattori M, Heilig H, Luna RA, van Hylckama Vlieg J, Junick J, Klymiuk I, Langella P, Le Chatelier E, Mai V, Manichanh C, Martin JC, Mery | 2017 | International Human Microbiome Standards | https://www.ebi.ac.uk/ena/data/view/PRJEB14847 | European Nucleotide Archive, PRJEB14847 |

C, Morita H,
O'Toole PW, Or-
vain C, Patil KR,
Penders J, Persson
S, Pons N, Popova
M, Salonen A,
Saulnier D, Scott
KP, Singh B, Slezak
K, Veiga P, Versa-
lovic J, Zhao L,
Zoetendal EG, Ehr-
lich SD, Dore J,
Bork P

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

## Appendix 1

DOI: https://doi.org/10.7554/eLife.46923.018

## Model details

### Qualitative or downstream effects of a step

In the most general formulation of our model first presented in the Results, we allow for the possibility that the protocol choice at a step influences the qualitative, as well as the quantitative, properties of the taxa. For instance, DNA extraction protocols yield differently fragmented DNA (*Costea et al., 2017*), which may affect the bias of downstream steps such as PCR and sequencing. Therefore, the bias at a step in general will depend on the protocol for that step as well as for all previous steps. To make this dependence explicit, let $P_l \mid P_1 \ldots P_{l-1}$ denote the $l$-th step of protocol $P$ performed after steps 1 through $l-1$ of protocol $P$. The bias of protocol $P$ is then

$$\mathbf{B}^{(P)} = \mathbf{B}^{(P_1)} \cdot \mathbf{B}^{(P_2 \mid P_1)} \cdot \mathbf{B}^{(P_3 \mid P_1 P_2)} \cdot \ldots \cdot \mathbf{B}^{(P_L \mid P_1 P_2 \cdots P_{L-1})}. \tag{14}$$

Our core assumption is that the bias at each step is fixed within the context of the total protocol $P$, but independent of the composition of the sample. All properties and implications of bias or differential bias of the total protocols described in the main text are valid under this condition in the presence of such qualitative or downstream effects.

Downstream effects are important to consider, however, when attempting to estimate the bias of an individual step or protocol choice. Consider using differential bias to estimate the effect of DNA extraction protocol in the *Costea et al. (2017)* experiment, where the protocols differ only at the extraction step. The differential bias between Protocols H and W, $\mathbf{B}^{(H/W)}$, includes both the direct effect of the differential bias at the extraction step as well as the indirect downstream effects the extraction protocol has on later steps. Here we cannot distinguish the two, as estimating the size of indirect effects requires experiments where protocols at different steps are varied in a combinatorial fashion.

A different but related issue arises in our estimation of bias from different steps in the *Brooks et al. (2015)* experiment. There, DNA extraction from pure cultures to create the DNA mixtures differed from extraction in the community samples (the cell mixtures). In order to estimate the bias during extraction, we must assume that there is no qualitative difference in the DNA, and so the bias of PCR, sequencing, etc. will be the same whether starting from the cell or the DNA mixtures. In other words, we must assume that the bias at a step $P_l$ is only a property of the individual step.

### Aggregates of taxa

Aggregating groups of taxa through the standard method of summing their abundances can cause bias to appear to vary among samples with different compositions. Consider four taxa with bias $(B_1, B_2, B_3, B_4)$ such that Taxon 1 is in Phylum 1, Taxon 2 is in Phylum 2, and Taxon 3 and Taxon 4 are both in Phylum 3. In a sample with actual composition $(A_1, A_2, A_3, A_4)$, the observed composition is $(A_1 B_1, A_2 B_2, A_3 B_3, A_4 B_4)$. If we aggregate taxa within phyla by the standard method, we observe the phylum-level composition $(A_1 B_1, A_2 B_2, A_3 B_3 + A_4 B_4)$. The bias we observe at the phylum level is

$$\left( \frac{A_1 B_1}{A_1}, \frac{A_2 B_2}{A_2}, \frac{A_3 B_3 + A_4 B_4}{A_3 + A_4} \right) = \left( B_1, B_2, \frac{A_3 B_3 + A_4 B_4}{A_3 + A_4} \right).$$

The observed efficiency of Phylum 1 relative to Phylum 2 remains $B_1/B_2$, independent of sample composition, but the efficiency of Phylum 3 relative to the other phyla depends on the relative abundances of Taxon 3 and Taxon 4.

The above example illustrates the general result that if we sum taxa that differ in their efficiencies, bias will no longer be independent of composition. As a result, estimates of bias at the aggregate level may find bias to vary among samples and perturbation-invariant analyses applied to taxonomic aggregates may not be invariant to bias.

When our measurement has sufficient taxonomic resolution to distinguish taxa at the level at which bias is conserved, then we can overcome these limitations by using the compositional approach to combining elements, which involves multiplying rather than summing the elements within each group (*van den Boogaart and Tolosana-Delgado, 2013*). For instance, if we multiply rather than sum taxa within a phylum we see that the error is again independent of composition,

$$\left(\frac{A_1 B_1}{A_1}, \frac{A_2 B_2}{A_2}, \frac{A_3 B_3 \cdot A_4 B_4}{A_3 \cdot A_4}\right) = (B_1, B_2, B_3 \cdot B_4).$$

Bias remains independent when individual taxa are replaced with products of taxa, and analyses that are perturbation invariant to bias on taxa will remain so when applied to taxa products. CoDA analyses that use this approach are typically based on *balances*, which are scaled and log-transformed ratios of the products of groups of taxa (*van den Boogaart and Tolosana-Delgado, 2013*). Applications of balances to microbiome analyses are described in *Silverman et al. (2017)* and *Rivera-Pinto et al. (2018)*.

## Appendix 2

DOI: https://doi.org/10.7554/eLife.46923.018

### Bias-estimation details

Bias estimation is complicated by the compositional nature of MGS measurement, particularly if different control samples contain different taxa. Because only relative abundances are measured, the measurement of a control sample $s$ only provides information about the efficiencies of the taxa in the sample *relative to each other*. If we restrict our scope to a set of $K$ possible taxa, then we can write the actual and observed compositions of $s$ as $K$-element compositions denoted $\mathbf{A}(s)$ and $\mathbf{O}(s)$. The measurement error can now be described by the compositional difference, $\mathbf{O}(s)/\mathbf{A}(s)$, which we denote $\mathbf{D}(s)$. Recall that the compositional difference is computed by element-wise division. If the sample $s$ contains all taxa—that is, if all of the elements of $\mathbf{A}(s)$ are positive—then $\mathbf{D}(s)$ is fully defined. But if $s$ does not contain all taxa, then some of the elements of $\mathbf{D}(s)$ are $0/0$ or undefined. The undefined elements of $\mathbf{D}(s)$ indicate that the control measurement tells us nothing about the bias between the taxa that are in $s$ and taxa that are not in $s$.

This appendix first summarizes concepts from the field of Compositional Data Analysis for computing the mean of a set of compositional vectors with undefined elements. These concepts allow us to equate the estimator of bias defined in **Equation 13** with the compositional mean, or *center*, of the compositional errors observed in the control samples,

$$\hat{\mathbf{B}} \sim cen\{\mathbf{D}(s)\}. \tag{15}$$

We then describe the condition for sufficient taxonomic overlap among the control samples for this estimate to be fully defined. Finally, we describe a bootstrap procedure for approximating the sampling distribution of the center that can be used to estimate uncertainty in the bias estimate.

### Distances and means in the compositional Aitchison geometry

The Aitchison geometry (**Pawlowsky-Glahn, 2015**; **Aitchison, 2003**, Chapter 2) defines a normed vector space for compositional vectors of a given length $K$, in which vector addition and subtraction are performed by element-wise multiplication and division and distance is measured according to the Aitchison norm. **Equation 11** shows that the Aitchison norm of $\mathbf{X}$ grows with the pairwise ratios among the elements or, equivalently, with the ratios of the elements to their geometric mean. The distance between two compositions $\mathbf{X}$ and $\mathbf{Y}$ in the Aitchison geometry is given by the Aitchison norm of their (compositional) difference and equals

$$\|\mathbf{X}/\mathbf{Y}\| = \sqrt{\frac{1}{K}\sum_{i<j}\left[\ln\frac{X_i/X_j}{Y_i/Y_j}\right]^2} = \sqrt{\sum_{i=1}^{K}\left[\ln\frac{X_i}{g(\mathbf{X})} - \ln\frac{Y_i}{g(\mathbf{Y})}\right]^2}. \tag{16}$$

*Equation 16* shows that the Aitchison distance between two compositions grows with the fold-difference in the element ratios between $\mathbf{X}$ and $\mathbf{Y}$.

The compositional mean, or *center*, of a set $\{X(s) \mid s \in S\}$ of compositional vectors with no undefined elements is typically defined by their element-wise geometric mean (**van den Boogaart and Tolosana-Delgado, 2013**, p. 74),

$$cen\{\mathbf{X}(s)\} \sim \left(\left[\prod_{s\in S}X_1(s)\right]^{1/|S|}, \ldots, \left[\prod_{s\in S}X_K(s)\right]^{1/|S|}\right), \tag{17}$$

where $|S|$ is the number of elements in the set $S$. Alternatively, we can define the center geometrically as the vector that minimizes the sum of squared distances between the compositions in $\{\mathbf{X}(s)\}$ and their center (*Pawlowsky-Glahn and Egozcue, 2002*),

$$\text{cen}\{\mathbf{X}(s)\} \sim \underset{\mathbf{Y}}{\arg\min} \sum_{s \in S} \|\mathbf{X}(s)/\mathbf{Y}\|^2. \tag{18}$$

In this way, the compositional mean can be seen to be directly analogous to the familiar mean in Euclidean geometry, which minimizes the sum of squared Euclidean distances between the vectors and their mean.

The geometric definition *Equation 18* can be used even when some of the compositions have elements that are undefined, provided that we first define the Aitchison distance between vectors with undefined elements. In the scenario we are interested in, the elements of $\mathbf{X} = \mathbf{D}(s)$ are undefined because the taxa corresponding to these elements were not present in the sample $s$, leaving us with no information about their ratios to the elements corresponding to the present taxa. It is therefore natural to compute the distance between $\mathbf{X}$ and a candidate estimate of the center $\mathbf{Y}$ with *Equation 16* restricted to just the defined elements. Equivalently, we can define the norm of $\mathbf{Z} \sim \mathbf{X}/\mathbf{Y}$ as the norm of the smallest-normed $K$-element vector that has the same ratios among the defined elements of $\mathbf{Z}$. This definition of the norm amounts to computing the norm of the $K^*$-element vector of the defined elements, as commented below *Equation 11*. To see why, note that the smallest $K$-element vector consistent with $\mathbf{Z}$ is given by setting the undefined elements of $\mathbf{Z}$ equal to the geometric mean of the defined elements. When we compute its norm using the second form in *Equation 11*, the logarithmic terms corresponding to these elements equal zero and so do not contribute to the sum.

This generalized definition of the norm now lets us use *Equation 18* to define the center of a set of compositional vectors that may have undefined elements. This definition can be used to compute the center numerically by minimizing the sum of squared distances. However, *van den Boogaart et al. (2006)* have developed an analytical approach based on projections of the log-transformed compositions; we refer readers to *van den Boogaart et al. (2006)* and *Bren et al. (2008)* for an explanation of this approach, which we use for all bias estimation in our analysis.

## Taxonomic overlap among control samples needed for a fully-determined bias estimate

Recall that the bias $\mathbf{B}$ is a compositional vector and so is defined only up to compositional equivalence, or an arbitrary positive constant multiplied by all elements. The bias estimate $\hat{\mathbf{B}}$ is uniquely determined (up to compositional equivalence) for all taxa present in at least one control sample only if there is sufficient taxonomic overlap among the samples. Here we explain the intuition for why this is the case and describe the general condition for sufficient overlap.

Suppose that there are two control samples; the first contains Taxon 1 and Taxon 2 and the second contains Taxon 3 and Taxon 4. Each sample allows us to learn the efficiencies of its taxa relative to others in that sample but not in the other sample. The first sample allows us to estimate $B_1/B_2$ and the second to estimate $B_3/B_4$, but we cannot estimate $B_1/B_3$. The situation improves if there is a third sample containing Taxon 2 and Taxon 4, allowing us to also estimate $B_2/B_4$. Now we can estimate $B_1/B_3$ by the relation $B_1/B_3 = (B_1/B_2)(B_2/B_4)(B_4/B_3)$.

With just the first two samples, the bias estimate $\hat{\mathbf{B}} = (\hat{B}_1, \hat{B}_2, \hat{B}_3, \hat{B}_4)$ is not well-defined. Because we have no information about the ratios between the two groups of taxa, we can always multiply the estimated efficiencies of one group (say, $\hat{B}_1$ and $\hat{B}_2$) by a constant $a$ while leaving the other group unchanged. The resulting distinct compositional vector would be just as consistent with the data from the control measurements in the sense that the residual sum of squared errors (RSS) would be unaffected. However, once we have the third sample, changing the ratios between the two groups of taxa in such a manner would increase the RSS

and yield a worse estimate of the bias. Thus, with the inclusion of the third sample there is a unique estimate $\hat{\mathbf{B}}$ that minimizes the RSS and the bias estimate is well-defined.

The general condition for determining if there is sufficient taxonomic overlap to uniquely determine $\hat{\mathbf{B}}$ can be stated using the concept from network theory of a network component. A (*connected*) *component* of an undirected network is a subnetwork whose nodes are connected by paths through the network and are not connected to nodes in any other component (*Newman, 2010*, p. 142). Define the *taxon co-occurrence network* as the network with taxa as nodes and undirected edges between any pair of taxa that co-occur in the same sample. Each component of the taxon co-occurrence network defines a set of taxa within which bias can be estimated, but for which the bias to other taxa cannot be estimated. Thus the estimate $\hat{\mathbf{B}}$ is uniquely determined if and only if the taxon co-occurrence network has a single component. When there are multiple components, the bias estimation method we have implemented (using the projection method of *van den Boogaart et al., 2006*) resolves ambiguities by giving the value of $\hat{\mathbf{B}}$ with the smallest Aitchison norm. The resulting estimate can still be used to analyze the bias as long as only the efficiency ratios between taxa that share a component are considered as having any meaning.

## Bootstrap method for estimating uncertainty in the bias

We describe two bootstrap (resampling) methods for estimating the uncertainty in the estimated bias. In the first method, based on the standard bootstrap introduced by *Efron (1979)*, the weights assigned to the control measurements in each resampling are drawn from a multinomial distribution. In the second method, based on the Bayesian bootstrap introduced by *Rubin (1981)*, the weights are drawn from a Dirichlet distribution. The two methods give approximately the same standard errors if all samples contain all taxa, but the second method avoids a pathology of the first when different samples contain different taxa.

Each method works by approximating the sampling distribution of $\hat{\mathbf{B}}$ by computing $\hat{\mathbf{B}}$ for a large number, $R$, of resamplings of the observed compositional errors, $\{\mathbf{D}(s) = \mathbf{O}(s)/\mathbf{A}(s)\}$. We first describe the multinomial resampling (standard bootstrap) method in the simple case where all control samples contain all taxa. In this method, a single bootstrap replicate consists of drawing $|S|$ error measurements by sampling with replacement from $\{\mathbf{D}(s)\}$ and computing the center of the resulting error measurements. Equivalently, we first draw a length-$|S|$ vector $\mathbf{w}$ from a $\mathrm{Multinomial}(|S|, (1/|S|, \ldots, 1/|S|))$ distribution to specify the weights assigned to the $|S|$ original measurements and compute the weighted center, $cen[\{\mathbf{D}(s)\}, \mathbf{w}]$, defined by

$$cen[\{\mathbf{X}(s)\}, \mathbf{w}] \sim \underset{\mathbf{Y}}{\arg\min} \sum_{s \in S} w_s \|\mathbf{X}(s)/\mathbf{Y}\|^2. \tag{19}$$

From each bootstrap replicate $r$ we obtain an estimate $\hat{\mathbf{B}}_r = cen[\{\mathbf{D}(s)\}, \mathbf{w}_r]$ of the bias that, because each sample contains all taxa, is necessarily a fully-determined $K$-element compositional vector.

We now use the approximate resampling distribution given by $\{\hat{\mathbf{B}}_r\}$ for a large number of replicates ($R$) to estimate the uncertainty in $\hat{\mathbf{B}}$. In order to respect the compositional nature of bias, we propose quantifying the uncertainty in $\hat{\mathbf{B}}$ by the geometric (multiplicative) standard errors in the relative efficiencies of various taxa. In the tables and figures, we typically present bias relative to the average taxon, which amounts to dividing the relative efficiency of each taxon by the geometric mean efficiency of all taxa. To estimate the geometric standard errors of the relative efficiencies in this case, we first geometrically center the elements of each $\hat{\mathbf{B}}_r$ (by taking $\hat{\mathbf{B}}_r \mapsto \hat{\mathbf{B}}_r / g(\hat{\mathbf{B}}_r)$) and then compute the element-wise geometric standard deviations over the $R$ replicates,

$$\text{gm. std. err. of } \frac{B_i}{g(\mathbf{B})} = \exp\sqrt{\frac{1}{R-1}\sum_{r=1}^{R}\left(\ln\frac{\hat{B}_{r,i}}{g(\hat{\mathbf{B}}_r)} - \frac{1}{R}\sum_{q=1}^{R}\ln\frac{\hat{B}_{q,i}}{g(\hat{\mathbf{B}}_q)}\right)^2}. \tag{20}$$

This standard error quantifies the uncertainty in the ratio of the efficiency of Taxon $i$ to the geometric mean efficiency of all taxa. Other ratios are also useful for understanding the uncertainty in the estimated bias. For example, one can estimate the standard error in the bias between taxa $i$ and $j$ by the geometric standard deviation of $\hat{B}_{r,i}/\hat{B}_{r,j}$ over the $R$ replicates.

The above procedure can fail if different samples have different taxa. Multinomial resamplings typically leave out some samples and thus can lack sufficient taxonomic overlap to give a fully-determined bias estimate even if $\hat{\mathbf{B}}$ is fully determined. A simple way to overcome this problem is to instead use Dirichlet resampling, as in the Bayesian bootstrap (*Rubin, 1981*). Procedurally, the Bayesian bootstrap proceeds like the weight-based formulation of the standard bootstrap given above, except that the weights $\mathbf{w}_r$ are sampled from a $\mathrm{Dirichlet}(1,\ldots,1)$ distribution rather than the multinomial distribution. Each sample now receives a positive weight (with probability 1), ensuring that each replicate estimate $\hat{\mathbf{B}}_r$ is fully determined so long as $\hat{\mathbf{B}}$ is. We can then proceed to compute compositional standard errors as before. The Dirichlet and multinomial weights of the two methods have the same expected value and approximately the same variance (*Rubin, 1981*). Consequently, the two methods will give approximately the same standard errors when all samples have all taxa.

To determine how much the precision of the bias estimate for Protocol H of *Costea et al. (2017)* varied with the number of control samples (*Figure 4—figure supplement 1*), we computed standard errors using the multinomial-weighted bootstrap method with the number of trials in the multinomial distribution equal to the specified number of control samples. That is, to estimate the precision associated with a given number $n$ of control samples ($1 \le n \le |S|$), we drew the length-$|S|$ weight vector $\mathbf{w}$ from a $\mathrm{Multinomial}(n, (1/|S|, \ldots, 1/|S|))$ distribution.

