## [Decision Letter]

Thank you for submitting your article "Consistent and correctable bias in metagenomic sequencing experiments" for consideration by *eLife*. Your article has been reviewed by three peer reviewers, including Peter Turnbaugh as the Reviewing Editor and Reviewer #1, and the evaluation has been overseen by Wendy Garrett as the Senior Editor. The following individuals involved in review of your submission have also agreed to reveal their identity: Christopher Quince (Reviewer #2); Sean Gibbons (Reviewer #3).

The reviewers have discussed the reviews with one another and the Reviewing Editor has drafted this decision to help you prepare a revised submission.

Summary:

The independent replication of microbiome research can produce very different results due to biases inherent in different protocols for studying complex microbial communities. Here, McLaren, Willis, and Callahan focus on multiplicative biases in microbiome experiments and quantify their effect on distorting the resulting abundance distributions. The authors formulate a rigorous framework in treating multiplicative biases based on individual protocol steps and derive a correction of that bias. Application of their correction methods to two mock community data sets validates their findings and provides improved estimates of microbial abundances. Together, this work provides an important framework to begin to more robustly compare data across experiments and research groups. This work is incredibly timely and provides a significant advance in our understanding of biases in microbiome data and a framework that will be built on to address them in the future.

Essential revisions:

1) The github repository is great but might not be enough of a tutorial for inexperienced users. I'd recommend writing a step by step tutorial (a la http://qiime.org/tutorials/).

Also, are there any plans in place to update this code as new approaches become available? It may also help to name the package/tool.

2) More discussion is needed regarding the propagation of noise or errors across multiplicative factors as opposed to bias. For instance, is noise in the extraction protocol more important than PCR because it occurs earlier? Or are the errors simply multiplicative too so that there is no purpose in trying to reduce noise in the earlier steps? In any case noise is perhaps more important than bias as it cannot be corrected so in their Tables 1 and 2 it would be good to have an estimate of the average error associated with each protocol. In some cases, a more biased but reproducible protocol could be preferable.

3) Both data sets used by the authors are derived from mock communities with known ground truth. The authors argue that their methods can be used even in the absence of a ground truth by using relative efficiencies. I feel that this is an important use case for real-world data sets. Standard communities, including all/most of the detected taxa across a set of real-world samples, are usually not available for any given microbiome study. Here, one could still estimate relative efficiencies by using technical replicates (samples co-processed across batches – how many samples need to be co-processed across batches for this approach to be useful?). However, that particular use case is discussed only briefly and there is no quantitative study with any data set that is not a mock data set. I feel that inclusion of a non-mock data set with technical replicates may be more informative here since that is closer to most published microbiome data sets. Based on that the authors could formulate a protocol that can be proposed for future microbiome experiments and that would allow systematic elimination of multiplicative biases. One challenge I foresee is that relative efficiencies will be very difficult to estimate for low-abundance taxa that are often unique to a small number of samples, unless every sample is processed across all batches.

4) The authors start by arguing that major biases should be multiplicative. Even though they provide a wide array of references describing the existence of bias in microbiome assays, there is no reference that supports the assumption that the strongest bias has to be multiplicative. This is addressed later in the manuscript by showing that multiplicative bias captures the error in mock communities well and by stating this multiplicative limitation in the Discussion. I would have liked a more systematic study of the potential magnitudes of multiplicative vs. additive/other biases.

a) For instance, what is the amount of variance explained by multiplicative bias vs. potentially additive biases (e.g. saturating effects, contamination from external sources, or well-to-well contamination)? I would expect that additive biases, like well-to-well contamination, to be more important in low-biomass samples. Further discussion of this point would be useful.

b) In the two data sets analyzed in this study the authors removed contamination manually. However, this approach likely won't be possible for non-mock data.

c) All bias estimates reported in the text are usually reported without dispersion measures, even though the formulated point estimate should induce an appropriate dispersion measure as well. It would be interesting to see how much uncertainty there is regarding the bias as a function of samples used for correction. This could aid in determining how many samples should be used to estimate the bias.

5) It would be worthwhile to discuss the observed bias structure with other data transformations (not only the ones from compositional data analysis). For instance, Equation 7 essentially makes fold changes of the same taxon across samples invariant to the bias. This would mean that the established transformation from RNA-seq analysis would also be applicable here (log-transformation and comparison by log-fold change rather than absolute difference).

6) I feel that some further investigation into correlations of the identified biases with taxonomy may be helpful to gain mechanistic insight into the sources of bias. For instance, does the bias correlate with the abundance of the taxon, or do phylogenetically similar taxa show similar biases? This could help to formulate strategies for estimating bias for taxa that are undetected in the batch replicates. The authors also argue that summarization of taxa into higher phylogenetic ranks may introduce additional errors, but they do not state which rank is permissible here (species, genus, family?).

The reviewers were all enthusiastic about publication and realize that all of these points may be difficult to accomplish within the next couple months. We would encourage the authors to use their best judgement in deciding what can be addressed in a substantive way and which points would be best addressed with additional discussion.

---

## [Author Response]

Essential revisions:1) The github repository is great but might not be enough of a tutorial for inexperienced users. I'd recommend writing a step by step tutorial (a la http://qiime.org/tutorials/).Also, are there any plans in place to update this code as new approaches become available? It may also help to name the package/tool.

We agree with the reviewers that a more accessible and better documented R package would be useful, and have now created a named package and a step-by-step tutorial.

We split off the generically useful functions into an R package called “metacal” (for “metagenomics calibration”), hosted on Github at https://github.com/mikemc/metacal. This package should be considered “alpha” but contains functions that can be used to measure bias in quality-control experiments that include samples of known composition. The GitHub Issues tracker will host discussions about the needs of potential users. We will update this package as we develop improved estimation and calibration methods.

We have added a tutorial to the R package that takes users through the process of estimating bias and performing calibration in the special case where the bias of all taxa can be directly measured from the control samples. The tutorial is viewable online at https://mikemc.github.io/metacal/articles/tutorial.html. We plan to expand upon this tutorial as we further our methods.

We have also taken several steps during the revision that make the package a more robust and useful tool for analyses beyond the present manuscript. In particular, we have:

Added functions to facilitate computing ratio-based statistics and diagnostic plots;Added a bootstrap procedure for estimating uncertainty in bias estimates (now described in Appendix 2 section “Bootstrap method for estimating uncertainty in the bias”);Implemented an analytical solution to the bias estimator for the case where different samples have different taxa, which is much faster than our previous optimization method and allows fast bootstrapping (described in Materials and methods section “Bias estimation” and motivated in Appendix 2 section “Distances and means in the compositional Aitchison geometry”);Added functions to check if there is sufficient taxonomic overlap among the control samples to obtain a fully determined bias estimate, or to find the disjoint groups of taxa within which bias can be estimated (fully explained in Appendix 2 section “Taxonomic overlap among control samples needed for a fully-determined bias estimate”).

Each of these features is demonstrated in the tutorial.

2) More discussion is needed regarding the propagation of noise or errors across multiplicative factors as opposed to bias. For instance, is noise in the extraction protocol more important than PCR because it occurs earlier? Or are the errors simply multiplicative too so that there is no purpose in trying to reduce noise in the earlier steps? In any case noise is perhaps more important than bias as it cannot be corrected so in their Tables 1 and 2 it would be good to have an estimate of the average error associated with each protocol. In some cases, a more biased but reproducible protocol could be preferable.

We have now added a measure of multiplicative noise to Tables 1 and 2 and described this measure and its application to the Costea et al. (2017) experiment in the second paragraph of the subsection “Bias measurement as a means of evaluating and improving protocols”. To allow direct comparison, we now use the analogous measure of bias instead of the previously used geometric standard deviation, although the new measure has a similar interpretation. In the Brooks et al. (2015) experiments (Table 1), we see the highest noise in the PCR mixtures, which is likely due to random error during sample construction resulting in imprecise estimates of the actual compositions, rather than noise during the MGS measurement. In the Costea et al. experiment (Table 2), the noise is smaller for protocols that preferentially measured the spike-in, suggesting that limited sequencing depth or other discrete-sampling effects may be having an effect.

Our quantification of noise assumes a multiplicative model of noise, and found that it gave reasonable results consistent with our expectations (e.g. noise << bias in most cases). However, we have not proven that all sources of noise in MGS measurements are multiplicative. Some noise mechanisms are likely to be well-modeled as multiplicative. For example, PCR amplification efficiencies may vary based on location of the PCR tube in the PCR machine, introducing random variation in a multiplicative factor (the efficiencies) that is independent of the sample’s composition. However, at least one source of noise is known to not be well-modeled as multiplicative: the random variation around the relative abundances introduced by the sequencing sampling process is larger and more asymmetric for rare taxa.

In general, we expect that noise will be reasonably modeled as multiplicative for reasonably abundant taxa. This means that, as the reviewers noted, the effect of multiplicative noise will be the same whether it arises in an early step or a late step. However, further investigation is warranted, and we expect at a minimum that a proper accounting for the non-multiplicative noise introduced by the generation of discrete sequencing reads will be necessary for robust bias estimation in low abundance taxa.

3) Both data sets used by the authors are derived from mock communities with known ground truth. The authors argue that their methods can be used even in the absence of a ground truth by using relative efficiencies. I feel that this is an important use case for real-world data sets. Standard communities, including all/most of the detected taxa across a set of real-world samples, are usually not available for any given microbiome study. Here, one could still estimate relative efficiencies by using technical replicates (samples co-processed across batches – how many samples need to be co-processed across batches for this approach to be useful?). However, that particular use case is discussed only briefly and there is no quantitative study with any data set that is not a mock data set. I feel that inclusion of a non-mock data set with technical replicates may be more informative here since that is closer to most published microbiome data sets. Based on that the authors could formulate a protocol that can be proposed for future microbiome experiments and that would allow systematic elimination of multiplicative biases. One challenge I foresee is that relative efficiencies will be very difficult to estimate for low-abundance taxa that are often unique to a small number of samples, unless every sample is processed across all batches.

We agree with the reviewers that the restriction of our analyses to mock communities is a major limitation of the present study. We are actively working towards demonstrating the utility of our methods on natural samples from other experiments, with the ultimate goal of formulating protocols for both mock and non-mock calibration in future microbiome experiments. However, we determined that a thorough investigation of differential bias in natural samples would require follow-up article(s) for reasons we outline below.

First, as the reviewers noted, most species are not present in most samples in real-world datasets, and many of the taxa that are present are at very low abundances. These complications make robust estimation of the relative efficiencies more difficult in natural samples than in controls. Second, consistent matching between the taxa identified by different protocols is challenging in natural samples in which many similar taxa may be present. Consider a set of *E. coli* and *Shigella flexneri* strains that are present in the real-world dataset. The way that a 16S rRNA protocol and a shotgun metagenomics protocol will divide these strains into taxonomic units will be different, and matching between the taxonomic units derived from different MGS protocols can be surprisingly challenging. This issue is avoided in control samples that contain collections of strains that are taxonomically well-separated. Third, most currently available datasets that could be used for investigating differential bias (i.e. that contain multiple samples measured by different MGS protocols) do not include technical replicate measurements, which are necessary to be able to distinguish inconsistent bias from contamination, random error, or sample mislabeling. For example, such technical replicate measurements of the same samples are not present in the Costea et al. (2017) dataset we analyzed, which otherwise would be a useful dataset for investigating differential bias in the natural fecal portion of the measured communities. We believe that the problems posed by noise, contamination, and taxonomic matching in real-world datasets can be overcome by more careful modeling and data curation, but require further substantial work and dataset development that would overly complicate the present manuscript.

Despite these challenges, we believe that differential bias is a non-obvious concept that is critical for understanding and developing methods to reduce technical disagreement between studies, and that it is therefore worth introducing in this manuscript. In order to better support its practical utility given the reviewers’ concerns, we expanded the illustration of differential bias in the revised manuscript: 1) We added the spike-in contaminant in the Costea et al. (2017) experiment to Figure 4B to illustrate how a consistent differential bias can be observed even when the true abundance of the taxon is unknown (subsection “Metagenomic sequencing of fecal samples with a spike-in mock community”, last paragraph). 2) We added estimates of differential bias between the Costea et al. (2017) protocols including the contaminant taxon to Table 2. While not equivalent to demonstration on a completely natural community, we believe this at least shows the utility of differential bias in the case where true abundances are not known.

4) The authors start by arguing that major biases should be multiplicative. Even though they provide a wide array of references describing the existence of bias in microbiome assays, there is no reference that supports the assumption that the strongest bias has to be multiplicative. This is addressed later in the manuscript by showing that multiplicative bias captures the error in mock communities well and by stating this multiplicative limitation in the Discussion. I would have liked a more systematic study of the potential magnitudes of multiplicative vs. additive/other biases.a) For instance, what is the amount of variance explained by multiplicative bias vs. potentially additive biases (e.g. saturating effects, contamination from external sources, or well-to-well contamination)? I would expect that additive biases, like well-to-well contamination, to be more important in low-biomass samples. Further discussion of this point would be useful.

Note: To be consistent with the manuscript, we use the term “bias” exclusively for the systematic error caused by differences in the measurement efficiency among taxa.

Our aim has been to determine how the error due to bias behaves and can be addressed. In particular, we do not consider error due to contamination and taxonomic misassignment, which can lead to measurements of taxa not actually in the sample and cannot be expressed in our simple multiplicative framework. These other sources of error may be more important than bias in some contexts and efforts to obtain accurate and reproducible metagenomics measurements must ultimately grapple with all three types of error. We have added a paragraph to the Discussion (subsection “Limitations and Next Steps”) to clarify the potential for contamination to limit effective application of the present work. In this response, we discuss further the role of contamination in obtaining quantitative measurements of community differences. We discuss practical solutions for dealing with contamination in our response to 4 b).

**Contamination:** Like bias, contamination (either from the lab environment or reagents or cross-contamination between samples) is a ubiquitous source of systematic error in metagenomics experiments. Whether bias or contamination is more important will depend on the experimental platform and system and the specific goals of the experiment. For example, contamination will increase in importance for sequencing platforms with high rates of index/barcode switching, for low biomass samples, and for downstream analyses that are sensitive to taxa found in very small proportions. The measures we propose for quantifying bias are challenged by the presence of contamination, as evidenced by the fact that an arbitrary amount of contamination from a taxon not in the sample yields an infinite multiplicative error. Future work that incorporates our bias model with models of contamination can lead to improved measurement of each error type and improved understanding of the joint error process in applications. We have summarized these points in the Discussion subsection “Limitations and Next Steps”.

**Saturation and other systematic variation in taxon efficiencies:** Saturation – where a taxon’s efficiency decreases with its relative or absolute abundance in the sample – is a special case of the relative efficiencies systematically depending on sample composition. The residual compositional errors, ϵ^(s)∼O(s)/(A(s)⋅B^), include systematic and random variation in the efficiencies and so place an upper bound on the strength of such effects. With the exception of the Brooks et al. (2015) PCR-product mixtures, the residuals are much smaller than the bias (Figure 3—figure supplement 1; Figure 4—figure supplement 1), and the large residuals in the PCR-product mixtures can plausibly be explained by random sample construction error. These observations indicate that saturation or other systematic variation in the bias is not significant in the experiments we analyzed. It is possible that saturation would become apparent in samples with more extreme variation in taxon abundances, we note one potential example of this (Thomas et al., 2016) in the Discussion and further testing is warranted. Our model can serve as a null model for future researchers to look for such effects in their own protocols; since it properly accounts for compositionality, deviations from our model can correctly be interpreted as efficiencies varying with sample composition. Protocol design should strive for consistent efficiencies, which is likely to be much more attainable than eliminating bias.

b) In the two data sets analyzed in this study the authors removed contamination manually. However, this approach likely won't be possible for non-mock data.

Removal of contamination is indeed more difficult in natural control samples where the true taxa present in the sample are not known a priori. However, removal of contamination using current practices – such as filtering out taxa below a minimum frequency above which out-of-sample contribution is assumed negligible or applying a specialized contamination-removal procedure (Davis et al., 2018; Larsson et al., 2018) – can be used instead of manual removal. Because our model and method for estimating bias consider only the ratios among taxa, they apply equally well to arbitrary subcompositions of taxa from each sample. Therefore, one can restrict to the taxa in each sample one is confident are not contaminants and proceed to estimate bias from these subcompositions with the methods we describe. Future Bayesian or maximum-likelihood methods that jointly model contamination and bias may make it possible to avoid the loss of information associated with current contaminant filtering approaches. We have summarized these points in the Discussion subsection “Limitations and Next Steps”.

c) All bias estimates reported in the text are usually reported without dispersion measures, even though the formulated point estimate should induce an appropriate dispersion measure as well. It would be interesting to see how much uncertainty there is regarding the bias as a function of samples used for correction. This could aid in determining how many samples should be used to estimate the bias.

We agree with the reviewers that applications of bias estimation should account for uncertainty in the estimated bias, and understanding of the structure of this uncertainty for different control-sample designs should inform experimental planning. Towards these ends, we have taken several steps in this revision.

We now implement a bootstrap procedure for estimating the uncertainty in the estimate of bias. This procedure is described in Appendix 2 section “Bootstrap method for estimating uncertainty in the bias”, implemented in the R package, and demonstrated in the tutorial. We are developing more sophisticated maximum-likelihood and Bayesian estimation procedures that we plan to add R package and tutorials in the future and that overcome some limitations of our bootstrap method (namely, not accounting for sequencing depth and potentially suboptimal performance when different control samples contain different taxa).

In the online R Markdown documents, we use the bootstrap procedure to estimate standard errors for all bias estimates reported in the manuscript. Due to the large sample size of these experiments, the standard errors are small relative to the estimated bias parameters and we exclude them from the tables for clarity. Standard errors for the estimated bias for the Costea et al. (2017) experiment are plotted in Figure 4—figure supplement 1 and for the estimated bias of individual workflow steps for the Brooks et al. (2015) experiment in Figure 6.

We also examined how the precision of the bias estimate varies with number of control samples for the protocol with an intermediate degree of noise (Protocol H) in the Costea et al. (2017) experiment. To estimate the (geometric) standard errors for N control samples (from N=1 to N=9), we performed multinomial bootstrap resampling of N samples and computed the geometric standard deviation of the resulting efficiency estimates. Figure 5—figure supplement 1 shows the resulting geometric standard errors as a function of N, alongside the observed compositions for help in interpretation. This figure and analysis is discussed in the subsection “Calibration”. When noise in the controls is much lower than bias, as it is here, measuring just 1 control sample gives a sufficiently accurate estimate to be useful in calibration. However, multiple samples are needed to allow estimating uncertainty and may be essential if “heavy-tail” random errors are common (such as those seen in two samples in the figure).

5) It would be worthwhile to discuss the observed bias structure with other data transformations (not only the ones from compositional data analysis). For instance, Equation 7 essentially makes fold changes of the same taxon across samples invariant to the bias. This would mean that the established transformation from RNA-seq analysis would also be applicable here (log-transformation and comparison by log-fold change rather than absolute difference).

Equation 6 and Equation 7 demonstrate that bias cancels out in analyses based on fold changes (FCs) in the ratios of taxa. However, bias does not cancel out in non-ratio based analyses due to the reasons stated in the section “Systematic error in taxon ratios, but not in taxon proportions, is independent of sample composition”. Here we present the general result for the error in the estimated proportional FC of a taxon between two samples, which provides a direct point of comparison to Equation 6 and Equation 7. From Equation 4, we have that the proportional FC of taxon i from sample t to sample s isPr[O(s)]iPr[O(t)]i=Pr[A(s)]i⋅Bi/∑jPr[A(s)]jBjPr[A(t)]i⋅Bi/∑jPr[A(t)]jBj=Pr[A(s)]iPr[A(t)]i⋅∑jPr[A(t)]jBj∑jPr[A(s)]jBj.

The bias partially cancels, but not fully, due to the dependence of the observed proportions on the sample mean efficiency (SME); the result is an error in the estimated proportional FC equal to the reciprocal of the FC in the SME. This can even results in a sign if the true proportional FC of a taxon is smaller than and opposing the FC in the SME. We give a numerical example of such a sign error for Taxon 3 in Figure 2 (subsection “Analyses based on fold-changes in taxon ratios are insensitive to bias, while analyses based on taxon proportions can give spurious results”, first paragraph) where the taxon is observed to have a proportional FC greater than 1 but in fact has a proportional FC that is less than 1. In a recent paper, Kevorkian et al. (2018) suggested that bias cancels when computing the proportional FC between samples. However, their derivation did not properly account for the compositional nature of MGS measurements, which causes the appearance of the sample mean efficiency in our Equation 4 but that is missing in their Equation 3.

Whether spurious associations (or missed true associations) are a serious worry in applications appears to us an open question. It is easy to come up with examples of spurious associations in a given taxon due to another taxon changing from high to low frequency between samples (such as Taxon 1 in Figure 2; see also Brooks, 2016, p. 338). Such large single-species changes might be uncommon in some ecosystems; but large phylum-level changes could similarly cause spurious results if species’ efficiencies are similar within the phylum. On the other hand, it is possible that the SME is fairly stable across samples. In this case, the partial canceling may be sufficient for accurate estimates of differential abundance. Because the observed FCs of all taxa are multiplied by the same error term, the observed log-FCs remain correlated with the actual log-FCs. A plot of observed vs. actual log-FCs will appear systematically shifted from the y = x line by the log-FC in the SME; but such a shift might often be obscured in real data by noise. This result could explain the high correlation and apparent lack of a systematic shift seen in D'Amore et al. (2016) Figure 7, which considers this relationship for the FCs in proportions between two DNA mock communities. Thus determining the conditions in which bias causes spurious taxonomic associations remains an important line of future work.

Other non-ratio-based transformations currently employed in RNA-seq and microbiome differential-abundance analysis, such as rank transformation and quantile normalization, face a similar problem. However, the practical implications of bias for such analyses again remain unclear and requires further consideration via theory and/or simulation, for which our model can provide a conceptual foundation.

6) I feel that some further investigation into correlations of the identified biases with taxonomy may be helpful to gain mechanistic insight into the sources of bias. For instance, does the bias correlate with the abundance of the taxon, or do phylogenetically similar taxa show similar biases? This could help to formulate strategies for estimating bias for taxa that are undetected in the batch replicates. The authors also argue that summarization of taxa into higher phylogenetic ranks may introduce additional errors, but they do not state which rank is permissible here (species, genus, family?).

We agree that understanding the extent to which bias is conserved among phylogenetically similar species is critical for understanding the causes and consequences of bias and for developing practical estimation and correction methods. The patterns we observe provide some anecdotal evidence about the nature of phylogenetic conservation of bias. For example, the variation in efficiencies seen among the five Firmicute species in the Costea et al. (2017) experiment shows that bias is not universally conserved at the phylum level and may pose a problem for analyses based on Phylum-level aggregation (as noted in the third paragraph of the Discussion). As further evidence, we visualized differential bias against phylogeny for the spike-in taxa in the Costea et al. (2017) experiment including the contaminant *Shigella flexneri*, (see Author response image 1). Author response image 1 shows that the three most closely related species – *S. flexneri, S. enterica*, and *Y. pseudotuberculosis –* have very similar relative efficiencies. These three taxa are all within the family *Enterobacteriaceae*. On the other hand, the next two most closely related species, *B. hansenii* and *C. saccharolyticum* in the family *Lachnospiraceae*, do not have similar relative efficiencies. Thus these results are ambiguous regarding how conserved bias is at the family level.

Our ability to draw firm conclusions about the phylogenetic conservation of bias is limited by the small number of taxa in these datasets. For instance, only two taxa in the Brooks et al. (2015) dataset, *L. iners* and *L. crispatus*, share a genus, while none of the spike-in taxa in the Costea et al. (2017) experiment do. Our future plans include using additional mock and non-mock datasets to estimate the phylogenetic conservation of bias more broadly, so as to give more concrete guidance on when bias is (or is not) likely to remain consistent when aggregating at a given taxonomic level.

Certain phenotypes are known to interact mechanistically with the MGS measurement process to introduce bias. In this revision, we improved our analysis of the relationship between bias and one such phenotype – 16S copy number (CN) – by performing a more rigorous estimation of CN (described in Materials and methods subsection “Data and analysis code availability” and shown in an R-markdown document online) and improving the figure showing the relationship between bias and CN (Figure 6—figure supplement 1). Previously we followed Brooks et al. (2015) in basing our estimates of 16S CN on RefSeq annotations for the given species. We have now revised our CN estimates for *A. vaginae* from 1 to 2 and for *L. iners* from 1 to 5 (Table 3). Previously 16S CN did not predict total bias or PCR bias, and that 16S CN correction actually made the estimated proportions worse than no correction in both cell and DNA mixtures. With these new numbers, we find that 16S CN variation explains a substantial amount of estimated PCR bias and that CN correction substantially improves the estimated proportions for the DNA mixtures. We thus substantially revised our previous conclusion that CN does not explain PCR bias in this experiment; however, it remains that CN explains at best a small fraction of total bias. These new results are reflected in the last paragraph of the Results subsection “Bias measurement as a means of evaluating and improving protocols” and reflected in the improved fit in the middle column of Figure 3—figure supplement 2. Investigation of other potential phenotypic predictors of bias is of interest, but we are again limited in our ability to systematically survey potentially relevant phenotypes by the limited number of taxa in our datasets.